# PROTOTYPE-BASED REGULARIZATION LEARNING FOR TEXT-VIDEO RETRIEVAL

## ABSTRACT

This work addresses the Intra-Inter Conflict (IIC) dilemma in text-video retrieval, *i.e*., (a) *intra-category variance*, refers to category-consistent instances that display substantial distributional disparity, and (b) *inter-category similarity*, where instances belonging to different categories exhibit distributional coupling. Through an analysis of the learned feature and recalled samples of current models, we posit this conflict stems from the appearance bias issue, *i.e*., the matching process is dominated by superficial semantics shared across samples, which undermines the contribution of discriminative semantics. To this end, we propose **Prototype-based Regularization Learning** (**PRL**), which regularizes the semantic boundaries of features through a set of prototypes, so as to maximally compel the model to learn compact and distinctive representations for text-video retrieval task. Specifically, PRL performs within- and cross-instance clustering in the embedding space to assign informative prototypes to instances with similar categories. Moreover, a **Prototype Discriminating Loss** (**PDL**) is proposed that makes semantically correlated instances self-organize around their respective prototype while maintaining separation across different ones, and a **Prototype Projection Loss** (**PPL**) is devised to align video and text features by adaptively projecting prototypes into a shared semantic manifold, thereby fostering cross-modal correspondence. Extensive experiments on five datasets demonstrate that the proposed model-agnostic strategy significantly boosts the performance of existing models, *e.g.,* improving TempMe, X-Pool, and CLIP4Clip by **+6.5%**, **+3.1%**, and **+5.0%** of SumR on the MSR-VTT dataset. Code available at: https://anonymous.4open.science/r/PRL-200D.

## 1 INTRODUCTION

Text-video retrieval (TVR) (Luo et al., 2022) has emerged as a fundamental task in multimodal learning, aiming to retrieve relevant video content given natural language queries and vice versa. In recent years, large vision-language pretraining models such as CLIP (Radford et al., 2021) have showcased significant cross-modal representation capabilities, achieving remarkable performance on various retrieval benchmarks. These models formulate the learning objective as a contrastive loss to align visual and textual modalities, establishing discriminative representations of instances like video and text, which has become a general paradigm in TVR systems.

Despite advances in this field, existing TVR methods remain fundamentally constrained by an often-overlooked Intra-Inter Conflict (IIC) dilemma, *i.e*., category-consistent instances exhibit distributional inconsistency, while category-divergent instances display unexpected entanglement. As demonstrated in Fig. 1 (left), the video distributions derived from the SOTA DiscoVLA (Shen et al., 2025), are heavily discrete and intersecting, *e.g.,* the videos with identical category ("drinking") exhibit distributional disparity while the videos with different categories ("drinking" *vs.* "pouring") are coupled. In addition, based on the category annotations provided in the MSR-VTT dataset, we conduct a quantitative analysis of cosine similarities, comparing intra-category and inter-category feature relationships. As observed in Fig. 1 (right, top), we find that semantically similar videos exhibit feature drift within the same category, while semantically dissimilar videos show confounding correlations across categories. The origin of the issue lies in the fact that differentiating ambiguous samples requires exploring their discriminative semantics, while existing methods tend to exploit superficial appearance bias to perform retrieval. For instance, as shown in Fig. 1 (right, bottom),

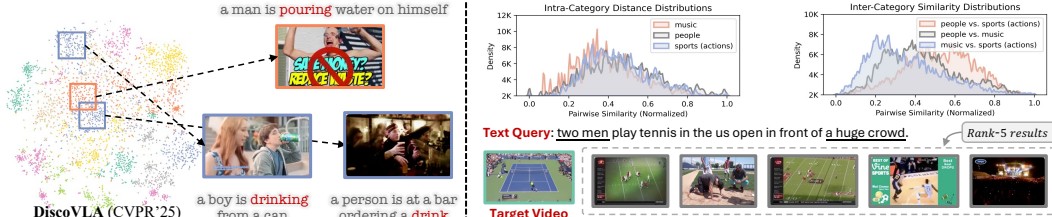

Figure 1: Intra-Inter Conflict (IIC) dilemma of learned distributions on the MSR-VTT dataset (Xu et al., 2016). **Left**: Instances belonging to the same category display inconsistent distributions, while those from different categories are unexpectedly entangled. **Right (top)**: Quantitative analysis further reveals that samples within the same category exhibit low semantic consistency, whereas samples from different categories show confounded correlations. **Right (bottom)**: The model tends to capture superficial semantics shared across samples (*e.g.,* visual backgrounds or common phrases such as "two men" or "a huge crowd") while neglecting discriminative semantics (*e.g.,* "tennis").

DiscoVLA is prone to recalling videos that contain a similar visual background to the target video or are dominated by frequent textual patterns in the text query, such as 'two men' or 'a huge crowd'.

Accordingly, we summarized the above issues as two challenges: (a) *Intra-category variance* and (b) *Inter-category similarity*. These two challenges motivate us to study the following problems: *i)* *"How to capture category-consistent semantics, thereby generating compact video-text representations within the same category?"*, and *ii)* *"How to differentiate those confounded representations of different categories for a more distinctive matching?"*. Our key intuition is that although video-text samples of the same category may vary in appearance, they can be aligned to category-level representation. By using such a representation as anchor, they will exhibit reduced variations within their categories in latent space, as well as provide stronger discriminative power between different ones.

Inspired by these observations, we propose a **Prototype-based Regularization Learning** (**PRL**) framework for text–video retrieval. PRL encourages the discovery of category-specific boundaries in the embedding space through a Clustering-based Prototype Mining strategy, which preserves compact representations for video/text instances within the same category while maintaining discrimination across different categories. Based on these established prototypes, we design a Prototype Discriminating Loss (PDL) that clusters semantically related instances around their respective prototypes while keeping different categories well-separated. Further, a Prototype Projection Loss (PPL) is introduced to align video and text features by projecting prototypes into a shared semantic space to promote cross-modal correspondence.

**Flexibility and Effectiveness.** Our PRL framework is orthogonal to existing TVR methods, hence it can be flexibly used to overcome the IIC dilemma for them. We evaluate our PRL using a broad spectrum of baseline methods, including CLIP4CLip (Luo et al., 2022), X-CLIP (Ma et al., 2022), X-Pool (Gorti et al., 2022), DiCoSA (Jin et al., 2023), DiscoVLA (Shen et al., 2025), and TempMe (Shen et al.) under two types of backbones (*i.e.*, CLIP-ViT-B/32 and CLIP-ViT-B/16). Experimental results on five datasets show that PRL consistently improves the performance of those methods, demonstrating the strong flexibility and effectiveness of our method. Notably, PRL enhances the baselines with trivial computational costs involved.

**Contributions.** Our main contributions are threefold. **1**) We provide an insightful analysis of the IIC dilemma in text-video retrieval, revealing that the root cause stems from the appearance bias issue. **2**) We propose the PRL framework to tackle the IIC dilemma from a prototype perspective, and PRL is orthogonal to existing text-video retrieval methods. **3**) Our PRL consistently improves the performance of the current TVR models on MSR-VTT, MSVD, DiDeMo, LSMDC, and ActivityNet datasets. Notably, by integrating into baseline models (TempMe, X-Pool, and CLIP4Clip), PRL achieves absolute gains of **3.1%~6.5%** in SumR on MSR-VTT dataset.

## 2 METHOD

In this section, we elaborate on the proposed method dubbed Prototype-based Regularization Learning (PRL) for text-video retrieval. The overall framework is illustrated in Fig. 2.

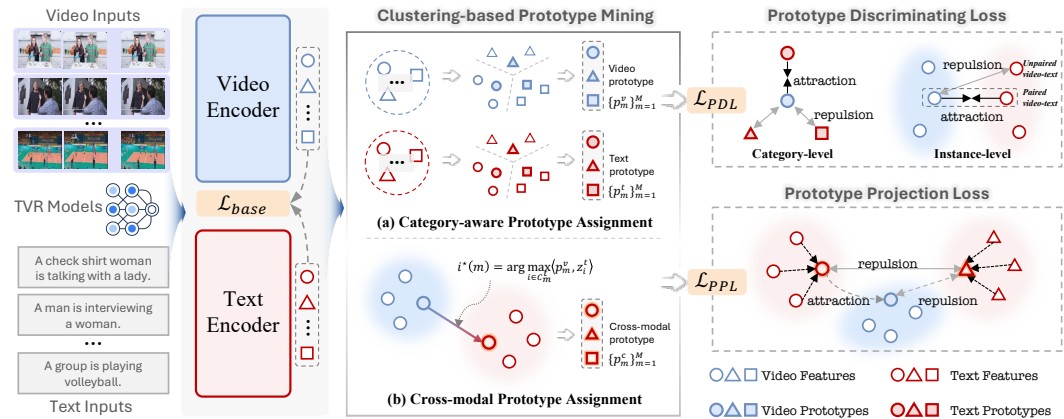

Figure 2: **Illustration of the proposed PRL.** During mini-batch training, the encoded video and text embeddings are processed by a clustering-based prototype mining module. Specifically, it comprises two main components: (a) Category-aware Prototype Assignment, which constructs informative video and text prototypes to cluster category-consistent video and text embeddings, and (b) Cross-modal Prototype Assignment, which derives cross-modal prototypes by searching for text features nearest to the video prototypes within paired clusters. Moreover, PRL imposes two constraints, *i.e.*, Prototype Discriminating Loss (PDL) and Prototype Projection Loss (PPL), which leverage the prototypes to produce compact video–text representations within the same category, while simultaneously enhancing the discrimination of confounded samples from different categories.

## 2.1 PROBLEM FORMULATION

Given a dataset of $N$ paired video-text instances $\{(v_i, t_i)\}_{i=1}^N$, where $v_i$ denotes a video and $t_i$ is its associated textual description, the goal of text-video retrieval is to construct a joint embedding space in which semantically aligned pairs are placed close together while unrelated pairs are pushed farther apart. This is achieved by introducing two modality-specific encoders: a video encoder $\phi_v(\cdot)$ and a text encoder $\phi_t(\cdot)$, which project the raw inputs into a $D$-dimensional embedding space. The semantic correspondence between a video $v_j$ and a text $t_i$ is typically evaluated using cosine similarity:

$$s_{ij} = \frac{\phi_t(t_i)^\top \phi_v(v_j)}{\|\phi_t(t_i)\| \cdot \|\phi_v(v_j)\|}, \tag{1}$$

where we denote $z_i^v = \phi_v(v_i)$, $z_i^t = \phi_t(t_i) \in \mathbb{R}^D$ as the encoded video and text feature representations respectively. The encoders are optimized through contrastive learning. In detail, for each aligned pair $(z_i^v, z_i^t)$, the objective is to maximize their similarity while simultaneously minimizing the similarity to all mismatched samples within the training batch:

$$\mathcal{L}_{\text{base}} = \frac{1}{2B} \left( \sum_{i=1}^B -\log \frac{\exp\left(\langle z_i^v, z_i^t \rangle / \tau\right)}{\sum_{j=1}^B \exp\left(\langle z_i^v, z_j^t \rangle / \tau\right)} + \sum_{i=1}^B -\log \frac{\exp\left(\langle z_i^t, z_i^v \rangle / \tau\right)}{\sum_{j=1}^B \exp\left(\langle z_i^t, z_j^v \rangle / \tau\right)} \right), \tag{2}$$

where $B$ means the mini-batch size, $\tau$ is a learnable temperature factor, and $\langle z_i^v, z_i^t \rangle$ refers to the similarity between video feature $z_i^v$ and text feature $z_i^t$. The optimization procedure enforces bidirectional alignment, *i.e.*, texts are encouraged to retrieve their corresponding videos and vice versa. At inference time, retrieval is carried out by ranking candidate instances of one modality according to similarity scores with a query from the other modality.

## 2.2 CLUSTERING-BASED PROTOTYPE MINING

To alleviate appearance bias, we propose discovering category associations among text–video samples through clustering, which compels the model to learn compact and discriminative representations guided by the generated informative prototypes. To this end, we introduce a clustering-based prototype mining strategy, which comprises two components: (1) Category-aware Prototype Assignment and (2) Cross-modal Prototype Assignment. Specifically, the former builds representative

prototypes to group video–text samples sharing the same category, thus promoting more compact category-level representations, the latter leverages these prototypes to locate the most relevant text features within paired clusters, thereby producing cross-modal prototypes that strengthen video–text correspondence. The details are presented below.

**Category-aware Prototype Assignment.** For each mini-batch $\mathcal{B} = \{(v_i, t_i)\}_{i=1}^{B}$, we leverage $M$ video prototypes $\mathcal{P}_v = \{p_m\}_{m=1}^{M}$ to represent diverse semantic patterns for the current video samples, where $p_m$ is the $m$-th video prototype. To obtain discriminative and informative prototypes, we perform K-means clustering among video representations. Specifically, let $\mathcal{Z}^v = \{z_i^v\}_{i=1}^{B} \in \mathbb{R}^{B \times d}$ denote the video representations of current mini-batch, where $z_i^v = \phi(v_i) \in \mathcal{R}^d$ denotes the $i$-th encoded video embedding. Based on the K-means clustering results $\mathcal{C}^v = \{C_m^v\}_{m=1}^{M}$, we define the cluster assignment matrix $\mathbf{A} \in {0,1}^{\{B \times M\}}$ such that $A_{im} = 1$ iff $i \in C_m^v$, the $m$-th cluster. Then the video prototypes $\mathcal{P}^v \in \mathbb{R}^{M \times d}$ are computed as:

$$\mathcal{P}^v = \mathbf{D}^{-1}\mathbf{A}^{\mathrm{T}}\mathcal{Z}^v,$$

where $\mathbf{D} = \mathrm{diag}(\mathbf{A}^{\top}\mathbf{1}) \in \mathbb{R}^{M \times M}$ stores the cluster sizes. On the text side, we adopt a simple way to assign prototypes, *i.e.*, for each cluster of videos obtained via K-means, we collect the corresponding text features that are paired with the videos in that cluster into a group. This finally constructs the text clusters $\mathcal{C}^t = \{C_m^t\}_{m=1}^{M}$. By reusing the cluster indicator matrix $\mathbf{A}$, we obtain the text prototypes $\mathcal{P}^t = \{p_m^t\}_{m=1}^{M}$. Overall, these two types of prototypes will serve as anchors for the subsequent prototype-based regularization learning.

**Cross-modal Prototype Assignment.** We further consider cross-modal correlations to better capture semantic alignment between video and text embeddings within each paired cluster $(C_m^v, C_m^t)$. Intuitively, text descriptions often refer to generic objects or scenes shared across videos, causing textual features to entangle with both paired and unpaired videos and weakening retrieval performance. Therefore, we focus on aligning text embeddings to video prototypes, deliberately ignoring alignment in the reverse direction from video to text. Specifically, for each paired cluster $(C_m^v, C_m^t)$, we compute the semantic similarity between the video prototype $p_m^v$ and all text features in $C_m^t$. The nearest text feature is then selected as the anchor, which is regarded as the cross-modal prototype of the $m$-th cluster, denoted as $p_m^c$:

$$i^\star(m) = \arg \max_{i \in C_m^t} \langle p_m^v, z_i^t \rangle, \qquad p_m^c = z_{i^\star(m)}^t.$$

During model training, we encourage the remaining text embeddings within the cluster $C_m^t$ to align with $p_m^c$, thereby implicitly enhancing their correlation to the corresponding video prototype. This process is deployed for all clusters to obtain the cross-modal prototypes $\mathcal{P}^c = \{p_m^c\}_{m=1}^{M}$. In subsequent training, these prototypes provide stronger supervision for video-text alignment within paired clusters, which compactify distributions while also sharpening boundaries of different clusters to maintain distinctiveness. Note that the above clustering and prototype assignment are recomputed from scratch for every batch, *i.e.*, prototypes ($\mathcal{P}^v, \mathcal{P}^t, \mathcal{P}^c$) and corresponding clusters ($\mathcal{C}^v$ and $\mathcal{C}^t$) are batch-dependent and decoupled across training iterations, preventing drift or accumulation of clustering errors. To prevent potential empty clusters or assignment ties during clustering, we adopt concise and robust remedies, *i.e.*, empty clusters are reinitialized with a randomly selected sample from the current batch, while ties are resolved deterministically by assigning the sample to the cluster with the smallest centroid index.

## 2.3 PROTOTYPE-BASED REPRESENTATIVE LEARNING

By performing category-aware prototype assignment separately and cross-modal prototype assignment jointly for the video and text embeddings, we obtain a set of prototypes, *i.e.*, video prototypes $\mathcal{P}^v = \{p_m^v\}_{m=1}^{M}$, text prototypes $\mathcal{P}^t = \{p_m^t\}_{m=1}^{M}$, and cross-modal prototypes $\mathcal{P}^c = \{p_m^c\}_{m=1}^{M}$. This prompts a key question: *"how can we ensure that the learned embeddings are both well-structured and discriminative, with clearly separated prototypes?"*. To achieve this, we propose two prototype-oriented constraints, *i.e.*, *Prototype Discriminating Loss* and *Prototype Projection Loss*, which fully exploit the compact and distinctive relationships of video-text instances in latent space.

**Prototype Discriminating Loss.** Based on the constructed video and text prototypes $\mathcal{P}^v$ and $\mathcal{P}^t$, we first distinguish prototypes by maximizing their distance to enhance the representative discrimination. Specifically, we consider the semantic distance constraint to encourage the $m$-th video

prototype $p_m^v$ close to its corresponding text prototype $p_m^t$ while pushing away the others. The corresponding training object is defined as:

$$\mathcal{L}_{\text{PDL\_P}}^{v \to t} = -\frac{1}{M} \sum_{m=1}^{M} \log \frac{\exp\left(\langle p_m^v, p_m^t \rangle / \tau\right)}{\exp\left(\langle p_m^v, p_m^t \rangle / \tau\right) + \sum_{\substack{j=1 \\ j \neq m}}^{M} \exp\left(\langle p_m^v, p_j^t \rangle / \tau\right)}. \tag{3}$$

where $\tau > 0$ is the temperature hyper-parameter. In this formula, the positive is treated as the matched video–text prototype pair $(p_m^v, p_m^t)$, while all remaining text prototypes $\{p_j^t \mid j \neq m\}$ in the batch are treated as negatives for the video prototype $p_m^v$. A similar loss is adopted at the text-to-video direction, thereby the overall loss is formulated as $\mathcal{L}_{\text{PDL\_P}} = \mathcal{L}_{\text{PDL\_P}}^{v \to t} + \mathcal{L}_{\text{PDL\_P}}^{t \to v}$. Moreover, as the prototype pairs $(p_m^v, p_m^t)$ become more separated in the embedding space, video and text samples in the same cluster share consistent semantics. Therefore, we then consider these samples as hard negatives between each other in the same cluster and employ contrastive learning:

$$\mathcal{L}_{\text{PDL\_C}}^{v \to t} = -\frac{1}{M} \sum_{m=1}^{M} \frac{1}{|C_m^v|} \sum_{z_i^v \in C_m^v} \log \frac{\exp\left(\langle z_i^v, z_i^t \rangle / \tau\right)}{\exp\left(\langle z_i^v, z_i^t \rangle / \tau\right) + \sum_{\substack{z_j^t \in C_m^t \\ j \neq i}} \exp\left(\langle z_i^v, z_j^t \rangle / \tau\right)}. \tag{4}$$

where the optimization process is summarized as: $\mathcal{L}_{\text{PDL\_C}} = \mathcal{L}_{\text{PDL\_C}}^{v \to t} + \mathcal{L}_{\text{PDL\_C}}^{t \to v}$. In this way, we maintain gained discriminatory power among video and text prototypes at the category-level and further enhance that among hard negative samples at the instance-level effectively.

**Prototype Projection Loss.** To enforce correlation within each text cluster and its paired video prototype, we introduce a prototype projection loss. The motivation derived from that textual description may exhibit inherent ambiguity or semantic drift as it might refer to general objects or scenes (*e.g.,* 'a person walking in a park', 'a car on the road') that are shared across different video contexts. This overlap can lead to text features being semantically entangled with the paired video and also unpaired videos, weakening the retrieval performance. Formally, given the constructed cross-modal prototypes $\mathcal{P}^c = \{p_m^c\}_{m=1}^M$, the following objective regularizes the text embeddings $z_i^t \in C_m^t$ within each cluster close to its corresponding prototype $p_m^c$ and repel others:

$$\mathcal{L}_{\text{PPL}} = -\frac{1}{\sum_m |C_m^t|} \sum_{m=1}^{M} \sum_{i \in C_m^t} \log \frac{\exp(\langle z_i^t, p_m^c \rangle / \tau)}{\exp(\langle z_i^t, p_m^c \rangle / \tau) + \sum_{\substack{k=1 \\ k \neq m}}^{M} \exp(\langle z_i^t, p_k^c \rangle / \tau)}. \tag{5}$$

For each text feature $z_i^t$ belonging to cluster $C_m^t$, the pair $(z_i^t, p_m^c)$ forms the positive sample, while all other prototypes $\{p_k^c \mid k \neq m\}$ serve as negative samples.

### 2.4 Training Details

The overall learning objective of PRL treats the contrastive loss $\mathcal{L}_{\text{base}}$ as the base loss function. Then, we combine it with our two proposed losses, *i.e.*, Prototype Discriminating Loss $\mathcal{L}_{\text{PDL}}$ ($\mathcal{L}_{\text{PDL\_P}}$ and $\mathcal{L}_{\text{PDL\_C}}$) and Prototype Projection Loss $\mathcal{L}_{\text{PPL}}$, as the final training objective to optimize the model:

$$\mathcal{L}_{\text{total}} = \mathcal{L}_{\text{base}} + \alpha \mathcal{L}_{\text{PDL\_P}} + \beta \mathcal{L}_{\text{PDL\_C}} + \gamma \mathcal{L}_{\text{PPL}}, \tag{6}$$

where $\alpha$, $\beta$ and $\gamma$ are the coefficients. For all loss objectives, we use softmax normalization over the available positives and negatives within the current batch. Note that during inference, we follow the standard retrieval process without involving the prototype regularization strategy.

## 3 Experiments

### 3.1 Experimental Settings

**Datasets.** We evaluate our approach on five widely used text-video retrieval benchmarks: (1) **MSR-VTT** (Xu et al., 2016): This dataset contains 10,000 video clips, each annotated with 20 descriptive sentences. Following the protocol in (Jin et al., 2023), we use 9,000 clips for training and the remaining 1,000 for testing. (2) **MSVD** (Chen & Dolan, 2011): Comprising 1,970 YouTube videos with approximately 40 captions per video, this dataset is split into 1,200 for training, 100 for validation, and 670 for testing. (3) **DiDeMo** (Anne Hendricks et al., 2017): This dataset includes 10,000

Table 1: **Performance comparison with state-of-the-art models *w/* or *w/o* our PRL on MSR-VTT dataset.** We report both text-to-video retrieval (left) and video-to-text retrieval (right) results. † indicates the reproduced results using public code of the corresponding paper for a fair comparison.

| Method | R@1↑ | R@5↑ | R@10↑ | MdR↓ | MnR↓ | R@1↑ | R@5↑ | R@10↑ | MdR↓ | MnR↓ | SumR↑ |
|---|---|---|---|---|---|---|---|---|---|---|---|
| *CLIP-ViT-B/32* | | | | | | | | | | | |
| CLIP4Clip†[Neurocom.'22] | 42.8 | 70.0 | 79.8 | 2.0 | 16.5 | 42.0 | 70.9 | 79.7 | 2.0 | 12.0 | 385.2 |
| **+PRL** (Ours) | **43.1** | **70.9** | **80.6** | **2.0** | **16.1** | **43.4** | 70.7 | **81.5** | **2.0** | **11.6** | **390.2** (+5.0) |
| X-CLIP[MM'22] | 46.1 | 73.0 | 83.1 | 2.0 | 13.2 | 46.8 | 73.3 | 84.0 | 2.0 | 9.1 | 406.3 |
| **+PRL** (Ours) | **47.0** | **73.3** | **83.8** | **2.0** | **14.1** | **47.1** | **73.4** | 83.1 | **2.0** | 10.1 | **407.6** (+1.3) |
| X-Pool[CVPR'22] | 46.9 | 72.8 | 82.2 | 2.0 | 14.3 | 44.4 | 73.3 | 84.0 | 2.0 | 9.0 | 403.6 |
| **+PRL** (Ours) | **47.0** | **73.3** | **82.5** | **2.0** | 14.4 | **45.9** | **73.8** | **84.2** | **2.0** | **8.7** | **406.7** (+3.1) |
| DiCoSA[IJCAI'23] | 47.5 | 74.7 | 83.8 | 2.0 | 13.2 | 46.7 | 75.2 | 84.3 | 2.0 | 8.9 | 412.2 |
| **+PRL** (Ours) | **49.1** | 74.7 | **84.1** | **2.0** | **13.1** | **47.0** | 74.6 | 83.7 | **2.0** | 9.3 | **413.2** (+1.0) |
| DiscoVLA[CVPR'25] | 47.0 | 73.0 | 82.8 | - | 14.1 | 47.7 | 73.6 | 83.6 | - | 10.0 | 407.7 |
| **+PRL** (Ours) | **47.4** | **73.1** | 82.4 | 2.0 | **14.0** | **48.0** | **73.8** | **83.7** | 2.0 | **9.9** | **408.4** (+0.7) |
| TempMe[ICLR'25] | 46.1 | 71.8 | 80.7 | - | 14.8 | 45.6 | 72.4 | 81.2 | - | 10.2 | 397.8 |
| **+PRL** (Ours) | **46.6** | **72.6** | **81.4** | 2.0 | **14.5** | **46.9** | **73.7** | **83.1** | 2.0 | **10.1** | **404.3** (+6.5) |
| *CLIP-ViT-B/16* | | | | | | | | | | | |
| DiscoVLA[CVPR'25] | 50.5 | 75.6 | 83.8 | - | 12.1 | 49.2 | 76.0 | 84.7 | - | 8.6 | 419.8 |
| **+PRL** (Ours) | **50.7** | 75.6 | **84.0** | 2.0 | 12.4 | **49.6** | **76.4** | **85.5** | 2.0 | 9.2 | **421.8** (+2.0) |
| TempMe[ICLR'25] | 49.0 | 74.4 | 83.3 | - | 11.9 | 47.6 | 75.3 | 85.4 | - | 9.0 | 415.0 |
| **+PRL** (Ours) | **49.2** | **76.0** | **85.0** | 2.0 | **9.4** | **48.9** | **76.1** | 84.0 | 2.0 | **8.8** | **419.2** (+4.2) |

Table 2: **Performance comparison with state-of-the-art *w/* or *w/o* our PRL on the MSVD, DiDeMo, LSMDC, and ActivityNet datasets.** We report both text-to-video and video-to-text results, and all models employ only CLIP-ViT-B/32 without considering post-processing operations. † indicates the reproduced results using public code of the corresponding paper for a fair comparison.

| Method | MSVD | | | | DiDeMo | | | | LSMDC | | | | ActivityNet | | | |
|---|---|---|---|---|---|---|---|---|---|---|---|---|---|---|---|---|
| | R@1↑ | R@5↑ | R@10↑ | MnR↓ | R@1↑ | R@5↑ | R@10↑ | MnR↓ | R@1↑ | R@5↑ | R@10↑ | MnR↓ | R@1↑ | R@5↑ | R@10↑ | MnR↓ |
| *Text-to-Video Retrieval* | | | | | | | | | | | | | | | | |
| CLIP4Clip[Neurocom.'22] | 46.2 | 76.1 | 84.6 | 10.0 | 43.4 | 70.2 | 80.6 | 17.5 | 20.7 | 38.9 | 47.2 | 65.3 | 40.5 | 72.4 | - | 7.4 |
| **+PRL** (Ours) | **46.5** | **76.1** | **84.6** | 10.2 | **43.6** | **70.6** | 78.5 | **17.4** | **21.6** | **39.1** | **49.6** | **59.4** | 38.8 | **74.0** | 82.4 | **7.2** |
| X-CLIP†[MM'22] | 46.9 | 76.8 | 85.5 | 9.8 | 45.2 | 74.0 | - | 14.6 | 23.3 | 43.0 | - | 56.0 | 44.3 | 74.1 | - | 7.9 |
| **+PRL** (Ours) | **47.4** | **77.2** | **85.6** | 9.8 | **46.7** | **74.7** | 82.3 | **14.5** | **23.5** | 41.3 | 50.3 | **55.8** | **45.4** | **74.6** | 84.5 | 8.4 |
| DiCoSA[IJCAI'23] | - | - | - | - | 45.7 | 74.6 | 83.5 | 11.7 | 25.4 | 43.6 | 54.0 | 41.9 | 42.1 | 73.6 | 84.6 | 6.8 |
| **+PRL** (Ours) | - | - | - | - | **46.3** | **74.8** | 83.5 | 14.8 | **25.6** | **43.9** | 51.7 | 54.7 | **42.5** | 72.5 | 84.9 | **6.7** |
| TempMe[CVPR'25] | - | - | - | - | 48.0 | 72.4 | 81.8 | 13.7 | 23.5 | 41.7 | 51.8 | 53.5 | 44.9 | 75.2 | 85.5 | 6.9 |
| **+PRL** (Ours) | - | - | - | - | **48.4** | **72.6** | 81.9 | **13.6** | **24.1** | **42.0** | 50.9 | **50.7** | **45.0** | **75.9** | **86.1** | **6.8** |
| *Video-to-Text Retrieval* | | | | | | | | | | | | | - | - | - | - |
| CLIP4Clip[Neurocom.'22] | 56.6 | 79.7 | 84.3 | 7.6 | 42.5 | 70.6 | 80.2 | 11.6 | 20.6 | 39.4 | 47.5 | 56.7 | 42.5 | 74.1 | 85.8 | 6.6 |
| **+PRL** (Ours) | **61.2** | **84.8** | **90.3** | **5.0** | **42.8** | **70.9** | **80.5** | **11.4** | **21.2** | 37.8 | **48.2** | **54.0** | 40.5 | 72.5 | 83.9 | 7.5 |
| X-CLIP†[MM'22] | 60.8 | 86.2 | 91.2 | 4.2 | 43.1 | 72.2 | - | 10.9 | 22.5 | 42.2 | - | 50.7 | 43.9 | 73.9 | - | 7.6 |
| **+PRL** (Ours) | **62.3** | **87.5** | **92.5** | **4.1** | **44.8** | **73.1** | 82.8 | 14.4 | 19.8 | **42.3** | 48.3 | 54.5 | **44.1** | **74.7** | 85.6 | **7.5** |
| DiCoSA[IJCAI'23] | - | - | - | - | - | - | - | - | - | - | - | - | - | - | - | - |
| **+PRL** (Ours) | - | - | - | - | 45.9 | 74.6 | 83.4 | 11.1 | 22.2 | 41.0 | 49.6 | 51.4 | 39.3 | 70.5 | 83.4 | 7.7 |
| TempMe[CVPR'25] | - | - | - | - | 48.4 | 75.4 | 83.6 | 9.1 | 22.2 | 41.5 | 51.5 | 48.0 | 45.3 | 74.7 | 86.2 | 6.4 |
| **+PRL** (Ours) | - | - | - | - | **48.6** | **76.1** | **83.8** | **9.0** | **22.3** | 41.5 | 51.5 | **47.2** | **46.2** | **76.8** | **87.4** | **6.2** |

Flickr videos paired with a total of 40,000 captions. We follow the video-paragraph retrieval setup introduced in (Luo et al., 2022) for evaluation. (4) **LSMDC** (Rohrbach et al., 2015): Consisting of 118,081 short video clips, each paired with a single sentence, we adopt the split from (Gabeur et al., 2020) with 109,673 training, 7,408 validation, and 1,000 testing samples to ensure fair comparison. (5) **ActivityNet Captions** (Caba Heilbron et al., 2015): It contains approximately 20,000 videos across 200 human activity categories. We follow the setup of (Jin et al., 2023) for consistency.

**Evaluation Metrics.** Following standard practice in previous works (Jin et al., 2023; Luo et al., 2022; Gorti et al., 2022), we evaluate model performance using common text-video retrieval metrics: Recall at K (R@K, higher is better ↑), Median Rank (MdR, lower is better ↓), Mean Rank (MnR, lower is better ↓), and Sum of Recall (SumR, higher is better ↑). We report R@1, R@5, and R@10, where K is set to 1, 5, and 10, respectively.

**Implementation Details.** For each baseline method, we follow the original implementation settings (*e.g.,* feature backbone, batch size, and learning rate) and incorporate our PRL without altering model structures or parameters. For PRL, the number of prototypes during clustering is set to 16. The coefficients $\alpha, \beta$, and $\gamma$ are empirically set as 0.1, 0.12, and 0.01, respectively. The above hyperparameters are fixed across datasets.

## 3.2 COMPARISON WITH STATE-OF-THE-ARTS

**Results on MSR-VTT.** We evaluate the effectiveness of PRL by integrating it into six representative text–video retrieval baselines: CLIP4Clip (Luo et al., 2022), X-CLIP (Ma et al., 2022), X-Pool (Gorti et al., 2022), DiCoSA (Jin et al., 2023), DiscoVLA (Shen et al., 2025), and

Table 3: **Ablation study for the designed components of RPN on MSR-VTT dataset.** The baseline is TempMe (Shen et al.). "CL": Contrastive Loss $\mathcal{L}_{base}$ in Eq. (2).

| Setting | $\mathcal{L}_{PDL}$ | $\mathcal{L}_{PPL}$ | R@1↑ | R@5↑ | R@10↑ | MnR↓ | R@1↑ | R@5↑ | R@10↑ | MnR↓ | SumR↑ |
|---|---|---|---|---|---|---|---|---|---|---|---|
| ① CL only (**Baseline**) | ✗ | ✗ | 46.1 | 71.8 | 80.7 | 14.8 | 45.6 | 72.4 | 81.2 | 10.2 | 397.8 |
| ② CL+PDL (PDL=PDL_P+PDL_C) | ✓ | ✗ | **46.4** | **72.5** | **81.4** | **14.5** | **45.7** | **73.1** | **82.4** | **10.0** | 401.5 (+3.7) |
| *v1.* Only apply $\mathcal{L}_{PDL\_P}$ | ✓ | ✗ | 46.2 | 71.8 | 81.3 | 14.6 | 45.4 | 72.8 | 81.4 | 10.2 | 398.8 |
| *v2.* Only apply $\mathcal{L}_{PDL\_C}$ | ✓ | ✗ | 46.2 | 71.9 | 81.2 | 14.9 | 45.6 | 72.7 | 82.0 | 10.2 | 399.6 |
| ③ CL+PPL | ✗ | ✓ | 46.2 | 71.9 | 80.9 | 15.5 | 45.6 | 73.2 | 83.3 | 10.3 | 401.1 (+3.3) |
| ④ CL+PDL+PPL (**Our PRL**) | ✓ | ✓ | **46.6** | **72.6** | **81.4** | **14.5** | **46.9** | **73.7** | **83.1** | 10.3 | **404.3 (+6.5)** |

Table 4: **Ablation study of different prototype-based methods on MSR-VTT dataset.** The baseline model is adopted from (Li et al., 2023a) for a fair comparison.

| Method | R@1↑ | R@5↑ | R@10↑ | MdR↓ | MnR↓ | R@1↑ | R@5↑ | R@10↑ | MdR↓ | MnR↓ | SumR↑ |
|---|---|---|---|---|---|---|---|---|---|---|---|
| Baseline | 45.6 | 72.9 | 81.5 | 2.0 | 14.5 | 45.2 | 71.6 | 81.5 | 2.0 | 10.9 | 398.3 |
| PAU[NeurIPS'23] | 48.5 | 72.7 | **82.5** | 2.0 | 14.0 | 48.3 | **73.0** | 83.2 | 2.0 | 9.7 | 408.2 (+9.9) |
| **PRL** (Ours) | **48.9** | **73.0** | **82.5** | 2.0 | **13.8** | **48.5** | **73.0** | **83.6** | 2.0 | **9.4** | **409.4 (+11.1)** |

**TempMe** (Shen et al.). Experiments are conducted on the MSR-VTT dataset under two widely used backbones, CLIP-ViT-B/32 and CLIP-ViT-B/16, with results summarized in Tab. 1. Across all baselines, PRL consistently enhances performance on both text-to-video ($t2v$) and video-to-text ($v2t$) retrieval. Under CLIP-ViT-B/32, PRL yields substantial SumR improvements of +5.0%, +1.3%, +3.1%, +1.0%, +0.7%, and +6.5% over the six baselines, highlighting its effectiveness in strengthening video–text discrimination. When equipped with the stronger CLIP-ViT-B/16 backbone, PRL further boosts performance, demonstrating both robustness and scalability. These results provide compelling evidence of the general applicability and efficacy of our method in improving video–text representation learning.

**Results on other Datasets.** To further assess the robustness of our method, we report quantitative results on additional datasets, including MSVD, DiDeMo, LSMDC, and ActivityNet. Tab. 2 summarizes performance comparisons for both $t2v$ and $v2t$ retrieval tasks, where all models adopt the CLIP-ViT-B/32 backbone without post-processing. Overall, PRL achieves consistent improvements on MSVD and DiDeMo and yields comparable gains on LSMDC and ActivityNet. For example, on MSVD, PRL improves CLIP4Clip and X-CLIP by +0.3% and +0.5% in $t2v$, and by +4.6% and +1.5% in $v2t$ at R@1, respectively. Notably, PRL also delivers significant performance enhancements on DiDeMo across most baselines, verifying its ability to disentangle and discriminate confounded representations. Nevertheless, performance gains are less evident on the LSMDC and ActivityNet datasets. We attribute this to two factors: 1) both datasets contain a higher proportion of long-form videos with complex scenarios, which may require finer-grained modeling; and 2) the optimal hyperparameters of PRL may vary across datasets and baselines, whereas we employ a unified configuration for all experiments.

### 3.3 ABLATIONS

**Effectiveness of the Loss Design in PRL.** The proposed PRL consists of two key objectives: a prototype discriminating loss ($\mathcal{L}_{PDL}$) and a prototype projection loss ($\mathcal{L}_{PPL}$). We perform an ablation study on advanced TempMe (Shen et al.) by progressively integrating each component into the model. The experimental results are reported on the MSR-VTT dataset as shown in Tab. 3. From ① and ② in the table, we find that integrating $\mathcal{L}_{PDL}$ into the baseline model for model training yields clear gains. Specifically, applying only $\mathcal{L}_{PDL\_P}$ (*v1*) or $\mathcal{L}_{PDL\_C}$ (*v2*) already improves retrieval performance across all metrics, and combining them achieves further enhancements, demonstrating their complementary capabilities. Furthermore, as observed from ③, introducing $\mathcal{L}_{PPL}$ alone also provides substantial improvements on both $t2v$ and $v2t$ tasks, validating its effectiveness in boosting cross-modal alignment. Finally, integrating both $\mathcal{L}_{PDL}$ and $\mathcal{L}_{PPL}$

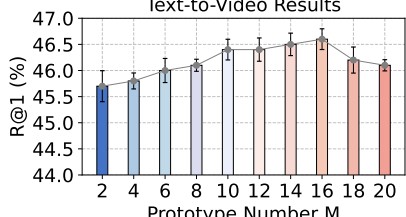

Figure 3: **Ablation study of the number of prototypes M.**

consistently achieves the best results as demonstrated in ④, delivering absolute improvements over the baseline method on all metrics and confirming the efficacy of the proposed PRL framework.

Table 5: **Ablation study of different prototype-based methods for cross-domain text-video retrieval.** We adopt baseline model from (Liu et al., 2019).

| Method | MSR-VTT $\rightarrow$ MSVD | | | | | | MSVD $\rightarrow$ MSR-VTT | | | | | |
| | Text-to-Image | | | Image-to-Text | | | Text-to-Image | | | Image-to-Text | | |
| | R@1↑ | R@10↑ | MnR↓ | R@1↑ | R@10↑ | MnR↓ | R@1↑ | R@10↑ | MnR↓ | R@1↑ | R@10↑ | MnR↓ |
|---|---|---|---|---|---|---|---|---|---|---|---|---|
| Baseline[BMVC'19] | 14.2 | 52.3 | 9 | 16.6 | 50.0 | 10 | 3.6 | 17.2 | 98 | 2.5 | 13.5 | 117 |
| ACP[CVPR'21] | 16.6 | 55.2 | 8 | 22.1 | 52.5 | 8 | 4.4 | 17.9 | 97 | 3.1 | 15.3 | 111 |
| **PRL** (Ours) | **17.8** | **56.9** | **7** | **23.4** | **53.4** | **7** | **6.5** | **18.5** | **92** | **4.9** | **16.8** | **104** |

**Comparison with other Prototype-based Methods.** We compare our PRL with other representative prototype-based approaches including PAU (Li et al., 2023a) and ACP (Liu et al., 2021) in traditional text-video retrieval and cross-domain text-video retrieval tasks. As observed in Tab. 4 and Tab. 5, PRL consistently outperforms competing methods across all evaluation metrics on two tasks. These results suggest that PRL is capable of learning cross-modal features that are both discriminative and transferable for effective retrieval. Additional experiments on MSCOCO for the text-image retrieval task are presented in the Appendix Tab. 9.

**Impact of the Prototype Number M.** To assess the number of prototypes required by the model, we conducted experiments with varying prototype numbers $M \in \{2, 4, 6, ..., 20\}$, as shown in Fig. 3. Our observations indicate that increasing $M$ allows the model to capture more latent semantic information. However, a continuous increase in prototype count does not always lead to consistent improvement. Notably, the model achieves its best performance when $M = 16$.

**Impact of the coefficients $\alpha$, $\beta$, and $\gamma$ on PRL.** We investigate the impact of coefficients in PRL, *i.e.*, $\alpha$ and $\beta$ in $\mathcal{L}_{PDL}$, as well as $\gamma$ in $\mathcal{L}_{PPL}$ in Eq. (6). We conduct experiments by varying the values of $\alpha$ and $\beta$ from 0 to 0.5 on the dataset MSR-VTT, and report the R@1 results in Fig. 4 (left). As can be observed, the performance of PRL gradually increases as the two parameters grow, after which the performance of PRL decreases. In particular, PRL establishes the best performance when $\alpha = 0.1$ and $\beta = 0.12$. Moreover, we also verify the impact of the coefficient $\gamma$ by setting different values ranging from 0 to 0.5, as illustrated in Fig. 4 (right). We find that the retrieval performance first improves before reaching the saturation point, *i.e.*, $\gamma = 0.01$, and then begins to fluctuate slightly. At last, we empirically choose the best values for $\alpha$, $\beta$, and $\gamma$ as 0.1, 0.12, and 0.01 in our final model.

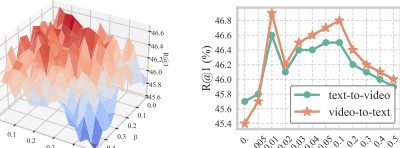

Figure 4: **Ablation study of coefficients $\alpha$, $\beta$, and $\gamma$ in Eq. (6).**

**Computational Cost.** This paper seeks to investigate cross-modal alignment via a plug-and-play manner. To this end, we present an analysis of the retrieval performance and complexity costs before and after integrating our PRL, encompassing training time, learnable parameters, GPU memory usage, and R@1 in Tab. 6. The

Table 6: **Computational costs of PRL on MSR-VTT dataset.** All models are tested on a NVIDIA RTX A6000 GPU with the same batch size.

| Method | Train Time↓ | # Params. ↓ | Memory↓ | R@1↑ (t2v) |
|---|---|---|---|---|
| CLIP4Clip | 372.34min | 123.54M | 13118MB | 42.8 |
| **+PRL** (Ours) | +1.54min | +0M | +3MB | **43.1** |
| TempMe | 19.28min | 0.5M | 11822MB | 46.1 |
| **+PRL** (Ours) | +0.59min | +0M | +3MB | **46.6** |

results show that PRL introduces only marginal training time increases, almost no additional parameters, and negligible GPU memory usage, while consistently improving R@1 performance. This highlights the efficiency of PRL as a lightweight and plug-and-play strategy.

## 3.4 PRACTICABILITY ANALYSIS OF PRL

To further verify the generalization ability of our method, we extend PRL to the image–text retrieval (ITR) task and multimodal embedding tasks. For ITR, we select five widely used baselines, including SGR (Diao et al., 2021), SAF (Diao et al., 2021), SGRAF (Diao et al., 2021), BiCro (Yang et al., 2023), and CLIP (Radford et al., 2021). We reproduce their reported results based on the official implementations and then directly integrate PRL into each model under the same configurations as in TVR. As shown in Tab. 7, PRL consistently improves the retrieval performance across all baselines. For instance, when applied to CLIP, one of the most representative cross-modal methods, PRL yields +1.6% and +2.2% gains in SumR on Flickr30K and MSCOCO, respectively. These results confirm that PRL not only enhances video–text retrieval but also exhibits strong generalization capacity to other retrieval tasks, highlighting its effectiveness as a versatile and plug-and-play module. For mul-

Table 7: **Generalization on image-text retrieval task on Flickr30K and MSCOCO datasets.**

| Method | Flickr30K (1K Test Set) | | | | | | | Method | MSCOCO (5K Test Set) | | | | | | |
|---|---|---|---|---|---|---|---|---|---|---|---|---|---|---|---|
| | Image-to-Text | | | Texte-to-Image | | | SumR↑ | | Image-to-Text | | | Text-to-Image | | | SumR↑ |
| | R@1↑ | R@5↑ | R@10↑ | R@1↑ | R@5↑ | R@10↑ | | | R@1↑ | R@5↑ | R@10↑ | R@1↑ | R@5↑ | R@10↑ | |
| SGR[AAAI'21] | 76.6 | 93.7 | 96.6 | 56.1 | 80.9 | 87.0 | 490.9 | SGR[AAAI'21] | 57.3 | 83.2 | 90.6 | 40.5 | 69.6 | 80.3 | 421.5 |
| +PRL (Ours) | 77.0 | 94.1 | 96.8 | 56.9 | 81.4 | 87.5 | 493.7(+2.8) | +PRL (Ours) | 58.1 | 84.0 | 91.2 | 41.4 | 70.5 | 81.0 | 426.2 (+4.7) |
| SAF[AAAI'21] | 75.6 | 92.7 | 96.9 | 56.5 | 82.0 | 88.4 | 492.1 | SAF[AAAI'21] | 55.5 | 83.8 | 91.8 | 40.1 | 69.7 | 80.4 | 421.3 |
| +PRL (Ours) | 75.9 | 93.0 | 97.2 | 56.4 | 82.8 | 89.1 | 494.4 (+2.3) | +PRL (Ours) | 56.4 | 83.6 | 91.8 | 40.8 | 70.6 | 81.8 | 425.0 (+3.7) |
| SGRAF[AAAI'21] | 78.4 | 94.6 | 97.5 | 58.2 | 83.0 | 89.1 | 500.8 | SGRAF[AAAI'21] | 58.8 | 84.8 | 92.1 | 41.6 | 70.9 | 81.5 | 429.7 |
| +PRL (Ours) | 78.6 | 94.4 | 97.9 | 58.7 | 83.8 | 89.4 | 502.8(+2.0) | +PRL (Ours) | 59.1 | 85.5 | 92.7 | 42.4 | 71.4 | 82.0 | 433.1 (+3.3) |
| BiCro[CVPR'23] | 61.6 | 90.4 | 96.0 | 79.1 | 96.4 | 98.6 | 522.1 | BiCro[CVPR'23] | 79.1 | 96.4 | 98.6 | 63.8 | 90.4 | 96.0 | 524.3 |
| +PRL (Ours) | 61.9 | 90.8 | 96.7 | 79.3 | 96.5 | 98.8 | 524.0(+1.9) | +PRL (Ours) | 79.8 | 96.8 | 98.8 | 64.0 | 90.9 | 96.1 | 526.4 (+2.1) |
| CLIPViT-B/32 | 78.7 | 95.4 | 98.0 | 66.3 | 88.6 | 93.1 | 520.0 | CLIPViT-B/32 | 56.3 | 81.7 | 89.4 | 42.8 | 71.2 | 81.1 | 422.6 |
| +PRL (Ours) | 78.9 | 95.8 | 98.1 | 66.4 | 89.0 | 93.4 | 521.6(+1.6) | +PRL (Ours) | 57.0 | 81.9 | 90.1 | 42.8 | 71.1 | 81.9 | 424.8 (+2.2) |

Table 8: **Generalization on the MMEB benchmark** (Jiang et al.). We average scores in each meta-task. "IND" is the in-distribution dataset, and "OOD" denotes the out-of-distribution dataset.

| Method | #Param. | Per Meta-Task Score | | | | Average Score | | |
|---|---|---|---|---|---|---|---|---|
| | | Classification | VQA | Retrieval | Grounding | IND | OOD | Overall |
| # Datasets→ | | 10 | 10 | 12 | 4 | 20 | 16 | 36 |
| VLM2Vec(Qwen2-VL) | 7B | 62.6 | 57.8 | 69.9 | 81.7 | 72.2 | 57.8 | 65.8 |
| **+PRL (Ours)** | | **63.2** | **57.9** | **70.0** | **82.3** | **72.8** | **58.5** | **66.1** |
| LLaVE | 0.5B | 57.4 | 50.3 | 59.8 | 82.9 | 64.7 | 52.0 | 59.1 |
| **+PRL (Ours)** | | **57.8** | **50.7** | 59.7 | **83.2** | **65.4** | 51.8 | **59.3** |
| LLaVE | 2B | 62.1 | 60.2 | 65.2 | 84.9 | 69.4 | 59.8 | 65.2 |
| **+PRL (Ours)** | | **62.9** | **60.9** | **66.0** | **85.4** | **70.2** | **60.1** | **65.9** |
| LLaVE | 7B | 65.7 | 65.4 | 70.9 | 91.9 | 75.0 | 64.4 | 70.3 |
| **+PRL (Ours)** | | **66.2** | **65.9** | **71.5** | 91.8 | **75.6** | **64.8** | **70.7** |
| UNITEinsruct | 2B | 63.2 | 55.9 | 65.4 | 75.6 | 65.8 | 60.1 | 63.3 |
| **+PRL (Ours)** | | **63.9** | **56.5** | **66.1** | **76.0** | **65.4** | **60.7** | **63.9** |
| UNITEinsruct | 7B | 68.3 | 65.1 | 71.6 | 84.8 | 73.6 | 66.3 | 70.3 |
| **+PRL (Ours)** | | **68.9** | **65.7** | **72.2** | **85.2** | **74.2** | **66.9** | **70.9** |

timodal embedding tasks, we integrate PRL into three mainstream models, *i.e.*, VLM2Vec (Jiang et al.), LLaVE (Lan et al., 2025), and UNITE (Kong et al., 2025), and evaluate them on the MMEB benchmark (Jiang et al.). As shown in Tab. 8, PRL consistently improves all methods across different parameter scales. For example, at the 7B level, PRL yields clear overall gains for VLM2Vec, LLaVE, and UNITEinstruct. These compelling results further verify the effectiveness and generality of our approach for multimodal embedding tasks.

### 3.5 QUALITATIVE ANALYSIS

**Visualization of Feature Distribution.** To intuitively illustrate the effectiveness of PRL in learning compact and discriminative representations, we employ t-SNE (Flexa et al., 2021) to visualize the feature distributions of DiscoVLA with and without PRL, as shown in Fig. 5. Without PRL (left), the features display considerable overlap across categories, indicating insufficient separability. In contrast, with PRL (right), the features form clearly separated clusters with enhanced intra-category compactness and inter-category distinctiveness. This visualization provides intuitive evidence of PRL's capability to produce more discriminative representations, thereby facilitating more effective retrieval.

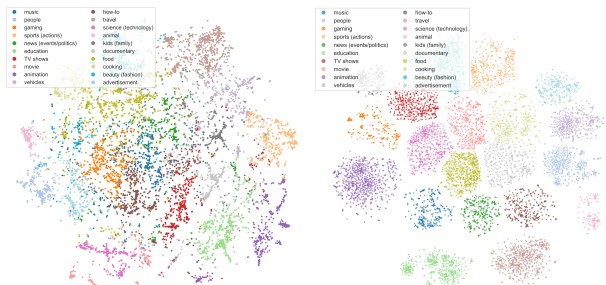

Figure 5: **t-SNE visualization of learned video features of DiscoVLA *w/o* and *w/* our PRL on MSR-VTT dataset.**

**Visualization of Retrieval Results.** To qualitatively validate the effectiveness of the proposed method, we provide text-to-video retrieval results as shown in Fig. 6. Specifically, we adopt the TempMe (Shen et al.) as a baseline model and integrate PRL for comparison. As observed, we find that TempMe *w/* PRL is capable of searching correct video, while TempMe is prone to confusion due to the visual scene (a green background) and the word "teams" within the text query. As a result, it returns a sample depicting a group of people engaged in an activity at the stadium, rather than specifically playing football. We observe similar properties in another example. These exam-

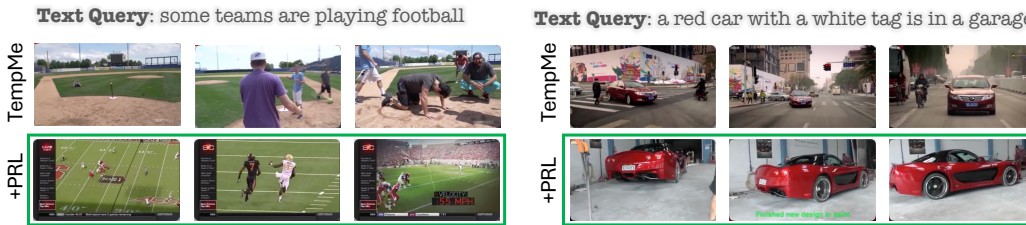

Figure 6: **Qualitative analysis of text-to-video results of TempMe (Shen et al.) *w/o* or *w/* our PRL.** Given the text query, we provide the R@1 results on MSR-VTT dataset. The correctly retrieved videos are highlighted in `green` box.

ples provide clear evidence that our method is capable of enhancing discrimination abilities both between different categories and within the same category.

## 4 RELATED WORK

**Text-Video Retrieval.** Text-video retrieval aims to rank videos according to their semantic relevance with textual queries, and vice versa. Early studies (Luo et al., 2022; Chen et al., 2020; Wang et al., 2025b;a; Xiao et al., 2025) mainly focused on designing fusion strategies to align pre-extracted textual and visual features, such as W2VV (Dong et al., 2018) and HGR (Chen et al., 2020). With the emergence of large-scale pre-trained image-text models like CLIP (Radford et al., 2021), remarkable progress has been made in image-text downstream tasks including classification and recognition, inspiring researchers to extend these models to video domain. For example, CLIP4Clip (Luo et al., 2022) enhances CLIP by incorporating temporal modeling, significantly improving video-text alignment and retrieval performance. X-Pool (Gorti et al., 2022) introduces a cross-modal attention that adaptively aggregates video frames conditioned on textual semantics, leading to more representative video embeddings. Despite progress, existing methods still struggle with the IIC dilemma, where intra-distribution dispersion and inter-distribution coupling degrade retrieval performance.

**Prototype Learning.** Prototype learning has garnered significant attention in the field of text-video retrieval (TVR) for its potential to enhance semantic alignment between textual queries and video content. Early works in this domain have explored various strategies to leverage prototypes for improved retrieval performance. Previous work (Lin et al., 2022) generates multiple visual prototypes per video with a variance loss to capture diverse semantics. (Li et al., 2023b) decomposes matching into object-phrase and event-sentence prototype phases, aligning spatial and temporal features. (Moon et al., 2025) encodes diverse video contexts into a fixed set of prototypes with cross- and uni-modal reconstruction tasks. (Zeng et al., 2021) introduces PAN, a prototype-based method that tackles unknown-category queries and modality imbalance to enhance the robustness of cross-modal retrieval. (Liu et al., 2021) proposes Adaptive Cross-Modal Prototypes, a method that enables cross-domain visual–text retrieval with unlabeled target data by preserving compositional structure and reducing distribution shift. (Li et al., 2023a) devises PAU that quantifies aleatoric uncertainty using evidential theory to provide more reliable cross-modal retrieval under low-quality or ambiguous data. However, prior works largely overlook the IIC dilemma, PRL explicitly introduces multiple prototypes to regularize the embedding space, strengthening intra-category cohesion and inter-category separation for more stable and robust video–text representations.

## 5 CONCLUSION

In this work, we address the intra–inter conflict dilemma in text–video retrieval, showing that appearance bias often drives models to capture superficial rather than discriminative semantics. To mitigate this issue, we propose PRL, a model-agnostic framework that constructs a structured embedding space with multiple discriminative prototypes, encouraging the learning of more robust representations. Moreover, two objectives are introduced—Prototype Discriminating Loss and Prototype Projection Loss—to regularize video–text embeddings, thereby improving intra-category compactness and inter-category separability. Extensive experiments demonstrate that PRL consistently boosts a variety of TVR models with negligible computational overhead, achieving state-of-the-art performance across multiple benchmarks.

## 6 THE USE OF LARGE LANGUAGE MODELS (LLMS)

During the preparation of this paper, Large Language Models (LLMs) are exclusively employed for the purpose of language refinement, which involves improving fluency, adopting academic expressions, and correcting minor grammatical errors.

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

## A  ADDITIONAL EMPIRICAL EVIDENCE OF IIC DILEMMA FROM MORE DATASETS AND PRELIMINARY METHODS.

We conduct a comprehensive analysis of the IIC dilemma across both text–video and text–image retrieval tasks. For text–video retrieval, we evaluate four representative methods: (1) widely used baseline: CLIP4Clip (Luo et al., 2022) and X-Pool (Gorti et al., 2022), and (2) state-of-the-art models: DiscoVLA (Shen et al., 2025) and TempMe (Shen et al.), on five benchmark datasets: MSVD (Chen & Dolan, 2011), MSR-VTT (Xu et al., 2016), DiDeMo (Anne Hendricks et al., 2017), LSMDC (Rohrbach et al., 2015), and ActivityNet Captions (Caba Heilbron et al., 2015). Since MSVD, DiDeMo, and LSMDC lack explicit category labels, we randomly sampled 1,000 videos from each and hired 10 experts to annotate them into ten categories (*i.e.*, *Daily Activities, Sports, Vehicles, Animals, Food, Nature, Entertainment, Science, Medical, Emergency*). For ActivityNet, which contains 200 human-annotated classes, we randomly sampled ten categories (*Archery, Baking cookies, Ballet, Brushing teeth, Canoeing, Cleaning windows, Horseback riding, Painting, Swimming, Walking the dog*) for analysis. For the examination on text–image retrieval, we conduct experiment on Flickr30K (Plummer et al., 2015) and MSCOCO (Lin et al., 2014) using two mainstream models: CLIP (Radford et al., 2021) and BiCro (Yang et al., 2023). Similarly, we sample 1,000 images for Flickr30K and annotate them into 10 categories with 10 human annotators. We use 80 default labels for the analysis of MSCOCO dataset. Fig. 7 presents the detailed category definitions used during annotation, including the fine-grained subcategories provided to annotators for reference. As shown in Fig. 8 and Fig. 10, we consistently find that category-consistent instances still exhibit strong distributional inconsistencies, while instances that deviate from their categories display unexpected entanglement patterns across these datasets and methods. These findings demonstrate that the proposed Intra–Inter Conflict dilemma arises consistently across diverse models and datasets, indicating that it reflects a general and systematic issue in current retrieval frameworks.

**Video Category for Annotation**

**1. Human Daily Activities**
Daily actions, household chores, office work, leisure, and social interactions
**2. Sports & Physical Performance**
Various sports, competitions, training, and extreme sports
**3. Vehicles & Transportation**
Cars, trains, airplanes, ships, traffic flows, and driving scenes
**4. Animals & Wildlife**
Pet behavior, wildlife, animal interactions, and ecological environments
**5. Food, Cooking & Dining**
Food preparation, cooking processes, dining scenes, and kitchen activities
**6. Nature, Landscapes & Weather**
Mountains, rivers, lakes, seas, urban outdoors, weather changes, and natural phenomena
**7. Entertainment & Performing Arts**
Dance, musical performances, stage shows, and talk programs
**8. Science or Industrial Scenes**
Experimental procedures, machinery operations, manufacturing processes, and factory environments
**9. Medical or Laboratory Activities**
Medical treatment, rehabilitation, medical operations, and laboratory environments
**10. Emergency Situations**
Surveillance scenarios, emergency events, traffic accidents, and public area dynamics

Figure 7: **Video categories for dataset annotation.**

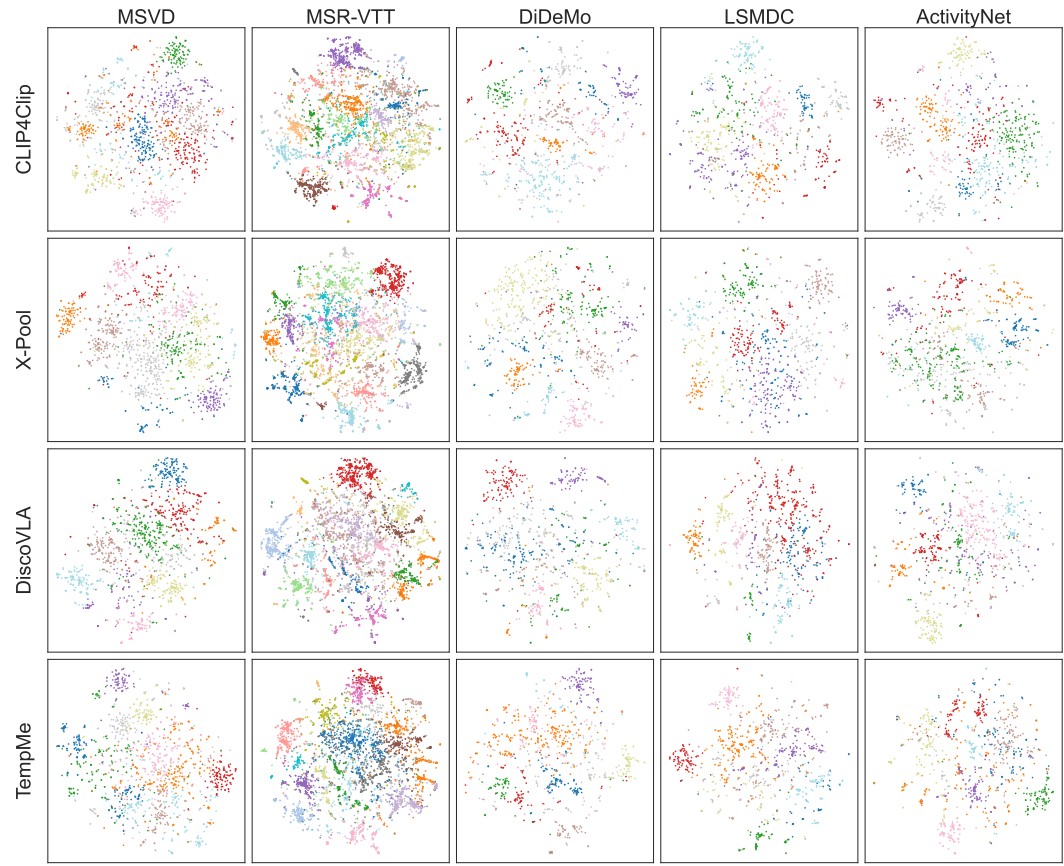

Figure 8: **Intra-Inter Conflict (IIC) dilemma of various methods in text-video retrieval task**, *i.e.*, CLIP4Clip, X-Pool, DiscoVLA, and TempMe on five widely used datasets: MSVD, MSR-VTT, DiDeMo, LSMDC, and ActivityNet.

# B   ADDITIONAL EXPERIMENTS

**Impact of Prototype Numbers across various Methods and Datasets.** In the main paper, we fix the number of prototypes to 16 across all methods and datasets for simplicity. Since the optimal choice of $M$ may depend on the characteristic of different methods and datasets, we additionally conduct experiments varying $M \in \{2, 4, 6, \ldots, 20\}$ for four methods: CLIP4Clip, X-Pool, DiscoVLA, and TempMe across five datasets: MSVD, MSR-VTT, DiDeMo, LSMDC, and ActivityNet Caption. In Fig. 12, we observed that larger datasets generally benefit from a higher number of prototypes, as this allows the model to capture more latent semantic diversity. However, increasing M beyond a reasonable range does not always lead to consistent performance gains. Excessive prototypes can fragment semantically similar instances across clusters and weaken the regularization effect, potentially reducing performance especially for datasets with small size. Notably, across most methods and datasets, the models achieve their best performance around $M$=16, indicating a balanced trade-off between capturing semantic richness and maintaining cluster coherence.

**Comparison with other Prototype-based Methods.**  We also compare our PRL with other prototype-based methods PAU on MSCOCO dataset for text-image retrieval task.  As shown in Tab. 9, PRL consistently outperforms the PAU across all metrics, showcasing its superiority.

**Impact of Different Clustering Algorithms for PRL.** In Tab. 10, we investigate the effect of different clustering algorithms: K-means, GMM, and DBSCAN, within PRL on three baseline methods including CLIP4Clip, DiscoVLA, and TempMe. Overall, PRL consistently improves retrieval performance across all baselines under different clustering algorithms, suggesting that PRL is robust to the choice of clustering strategy. These results demonstrate that PRL effectively enhances text-video

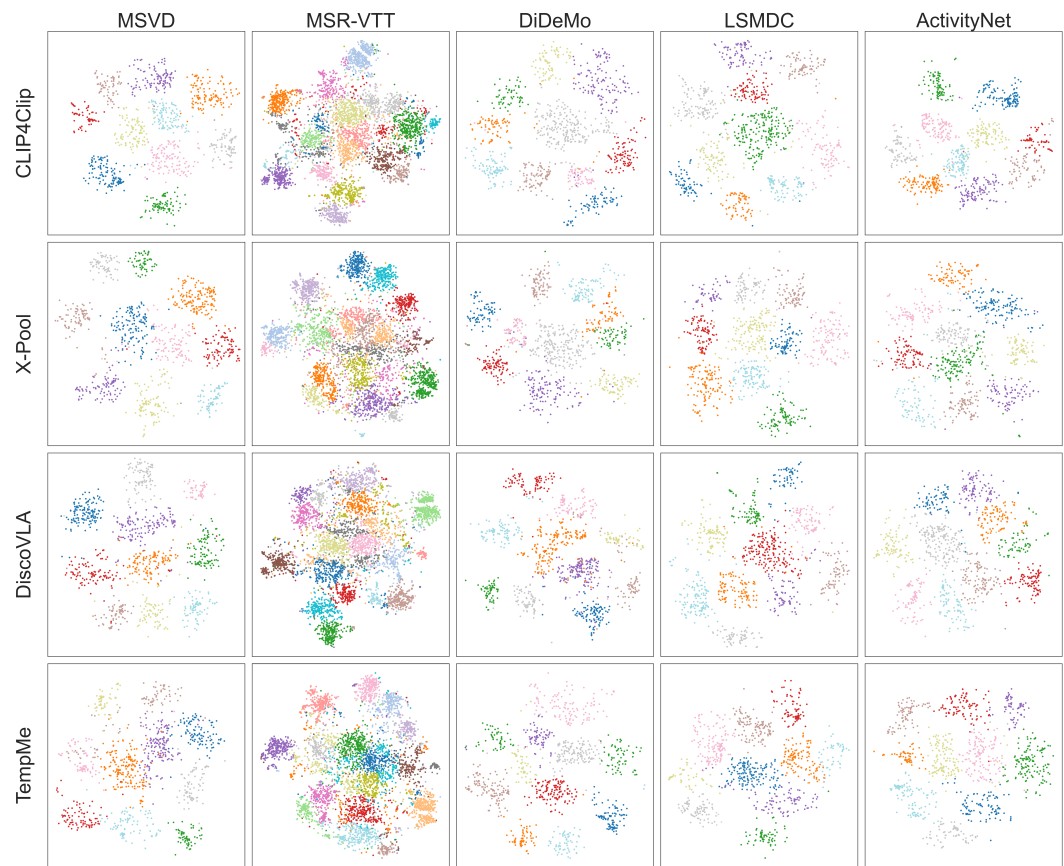

Figure 9: **t-SNE visualization of learned video features of various methods *w/* PRL on different text-video retrieval datasets.** For clearer insights, this figure is best viewed in comparison with Fig. 8.

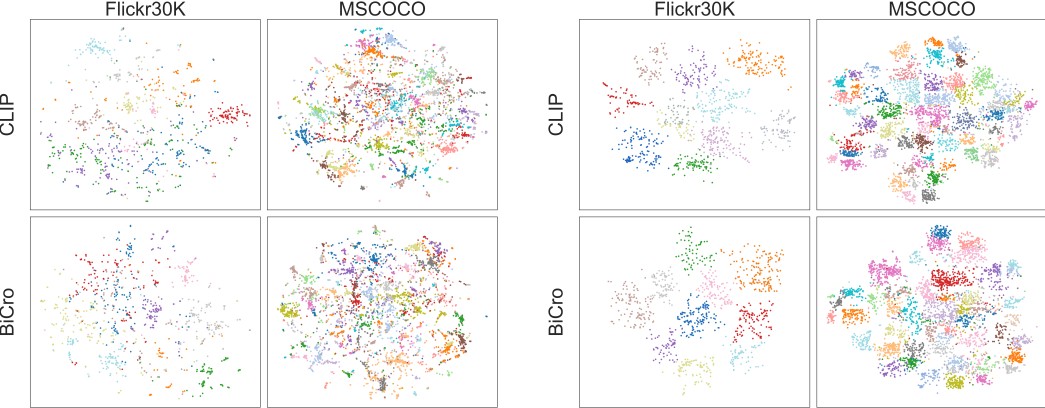

Figure 10: **Intra-Inter Conflict (IIC) dilemma of various methods in text-image retrieval task**, *i.e.*, CLIP and BiCro on two widely used datasets: Flickr30K and MSCOCO.

Figure 11: **t-SNE visualization of learned image features of various methods *w/* PRL.** For clearer insights, this figure is best viewed in comparison with Fig. 10.

retrieval by refining ranking, with modest sensitivity to the clustering algorithm used. For method simplicity and high usability, we adopt the simple yet effective K-means in PRL.

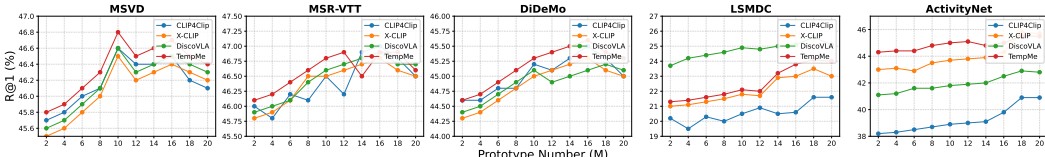

Figure 12: **Analysis of prototype numbers across different methods and datasets.**

**Impact of Different Batch Size during Clustering.** In the proposed PRL, we employ batch-level clustering to obtain the prototypes. It is necessary to investigate the impact of batch size $B$ on the performance of PRL. To this end, we set $B$ to the values of $\{64, 128, 256, 512\}$, and report the results of the TempMe *w/o* and *w/* our PRL on the MSR-VTT dataset in Tab. 11. As observed, our PRL consistently enhances the baseline across all batch sizes, demonstrating its robustness. Notably, the gains are more pronounced with larger batches, as the increased number of samples enriches semantic information, improving cluster and prototype discriminability and reinforcing the regularization effect.

**Impact of Category for Clustering.** In PRL, our default design relies on unsupervised K-means. To examine whether external semantic cues (*i.e.*, annotated video category labels) can further improve prototype quality, we additionally perform clustering using ground-truth categories on MSR-VTT. As shown in Tab. 12, category-guided clustering yields better performance (*e.g.,* SumR: 404.3 → 406.3), as explicit labels provide a stronger supervisory signal for grouping. However, this clustering strategy requires datasets with reliable category annotations, which are often costly and labor-intensive to obtain. PRL therefore adopts the unsupervised formulation to strike an effective balance between performance and annotation cost.

Table 9: **Ablation study of different prototype-based methods on MSCOCO dataset.** The baseline model is adopted from (Li et al., 2023a) for a fair comparison.

| Method | MSCOCO (1K Test Set) | | | | | | MSCOCO (5K Test Set) | | | | | |
|---|---|---|---|---|---|---|---|---|---|---|---|---|
| | Image-to-Text | | | Text-to-Image | | | Image-to-Text | | | Text-to-Image | | |
| | R@1↑ | R@5↑ | R@10↑ | R@1↑ | R@5↑ | R@10↑ | R@1↑ | R@5↑ | R@10↑ | R@1↑ | R@5↑ | R@10↑ |
| Baseline | 80.1 | 95.7 | 98.2 | 67.1 | 91.4 | 96.6 | 62.9 | 84.9 | 91.6 | 46.5 | 73.8 | 82.9 |
| PAU[NeurIPS'23] | 80.4 | 96.2 | 98.5 | 67.7 | 91.8 | 96.6 | 63.6 | 85.2 | 92.2 | 46.8 | **74.4** | 83.7 |
| **PRL** (Ours) | **80.9** | **96.7** | **98.6** | **67.8** | **92.0** | **96.9** | **63.8** | **85.5** | **92.4** | **46.9** | 74.4 | **84.0** |

Table 10: **Ablation study of different clustering algorithms for RPL.**

| Method | R@1↑ | R@5↑ | R@10↑ | MdR↓ | MnR↓ | R@1↑ | R@5↑ | R@10↑ | MdR↓ | MnR↓ |
|---|---|---|---|---|---|---|---|---|---|---|
| CLIP4Clip[Neurocom.'22] | 42.8 | 70.0 | 79.8 | 2.0 | 16.5 | 42.0 | 70.9 | 79.7 | 2.0 | 12.0 |
| **+PRL** (K-means) | **43.1** | **70.9** | **80.6** | 2.0 | **16.1** | **43.4** | 70.7 | 81.5 | 2.0 | 11.6 |
| **+PRL** (GMM) | 42.9 | 69.7 | 80.1 | 2.0 | 16.5 | 42.9 | 70.8 | **81.6** | 2.0 | 12.3 |
| **+PRL** (DBSCAN) | 43.0 | 70.2 | 80.4 | 2.0 | 16.3 | 43.2 | **71.0** | 81.0 | 2.0 | **11.3** |
| DiscoVLA[CVPR'25] | 47.0 | 73.0 | **82.8** | - | 14.1 | 47.7 | 73.6 | 83.6 | - | 10.0 |
| **+PRL** (K-means) | **47.4** | 73.1 | 82.4 | 2.0 | **14.0** | **48.0** | 73.8 | **83.7** | 2.0 | **9.3** |
| **+PRL** (GMM) | 47.2 | **73.2** | 82.5 | 2.0 | **14.0** | 47.3 | 73.8 | 83.5 | 2.0 | 9.9 |
| **+PRL** (DBSCAN) | **47.4** | 73.1 | 82.6 | 2.0 | 14.1 | 47.5 | 73.5 | 83.4 | 2.0 | 10.0 |
| TempMe[ICLR'25] | 46.1 | 71.8 | 80.7 | - | 14.8 | 45.6 | 72.4 | 81.2 | - | 10.2 |
| **+PRL**(K-means) | **46.6** | **72.6** | 81.4 | 2.0 | **14.5** | **46.9** | 73.7 | **83.1** | 2.0 | **10.1** |
| **+PRL** (GMM) | 46.1 | 72.3 | **81.7** | 2.0 | 14.7 | 45.7 | **74.0** | 82.0 | 2.0 | 10.2 |
| **+PRL** (DBSCAN) | 46.3 | 72.5 | 81.5 | 2.0 | 14.6 | 46.0 | 73.5 | 82.5 | 2.0 | 10.2 |

## C  ADDITIONAL QUALITATIVE ANALYSIS

**Visualization of Feature Distribution.** We further conduct qualitative analyses across multiple models and datasets for both text–video and text–image retrieval in Fig. 9 and Fig. 11. Compared to the methods without PRL Fig. 8 and Fig. 10, introducing PRL leads to clearer and more compact cluster structures, with substantially reduced overlap among semantically distinct groups. These

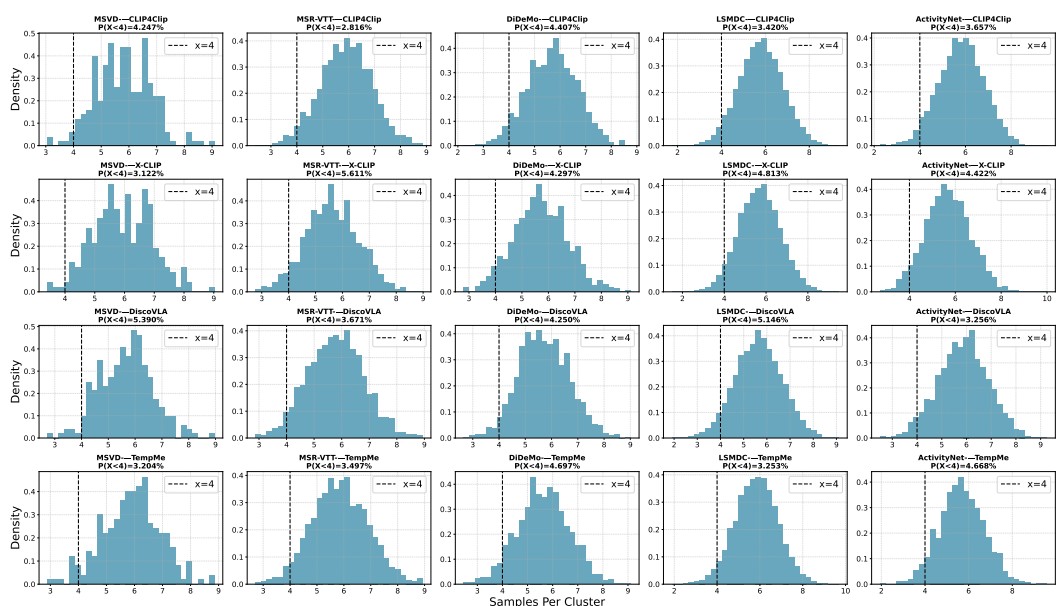

Figure 13: **Clustering stability analysis.**

Table 11: **Ablation study of different batch size (B) for PRL.**

| Method | R@1↑ | R@5↑ | R@10↑ | MdR↓ | MnR↓ | R@1↑ | R@5↑ | R@10↑ | MdR↓ | MnR↓ | SumR↑ |
|---|---|---|---|---|---|---|---|---|---|---|---|
| *B=64* | | | | | | | | | | | |
| TempMe | 45.1 | 71.0 | 80.0 | 2.0 | 15.8 | 45.4 | 72.0 | 81.5 | 2.0 | 11.7 | 395.0 |
| **+PRL** | **46.3** | **72.2** | **81.1** | 2.0 | **14.8** | **46.0** | **72.8** | **82.6** | 2.0 | **10.6** | **402.0** (+6.0) |
| *B=128* | | | | | | | | | | | |
| TempMe | 46.1 | 71.8 | 80.7 | - | 14.8 | 45.6 | 72.4 | 81.2 | - | 10.2 | 397.8 |
| **+PRL** | **46.6** | **72.6** | **81.4** | 2.0 | **14.5** | **46.9** | **73.7** | **83.1** | 2.0 | **10.1** | **404.3** (+6.5) |
| *B=256* | | | | | | | | | | | |
| TempMe | 46.7 | 72.5 | 81.2 | 2.0 | 14.5 | 46.8 | 73.5 | 82.0 | 2.0 | 10.0 | 402.7 |
| **+PRL** | **46.9** | **73.4** | **82.8** | 2.0 | **14.3** | **47.4** | **74.6** | **84.3** | 2.0 | **9.9** | **409.4** (+6.7) |
| *B=512* | | | | | | | | | | | |
| TempMe | 47.0 | 72.7 | 81.4 | 2.0 | 14.5 | 47.2 | 73.8 | 83.0 | 2.0 | 9.8 | 405.1 |
| **+PRL** | **47.5** | **73.9** | **82.6** | 2.0 | **14.4** | **48.7** | **74.4** | **85.0** | 2.0 | **9.6** | **412.1** (+7.0) |

qualitative patterns align closely with the observed quantitative gains and demonstrate that PRL effectively enhances intra-category compactness while improving inter-category separation. Overall, these results verify PRL's ability to shape a more discriminative and well-structured embedding space across diverse retrieval scenarios.

**Analysis of Clustering Stability.** To assess clustering stability in PRL, we evaluate the distribution of samples within each cluster across all batches during training in Fig. 13. Specifically, we select four representative models (CLIP4Clip, X-CLIP, DiscoVLA, and TempMe) and evaluate them on five datasets (MSVD, MSR-VTT, DiDeMo, LSMDC, and ActivityNet). As shown, the proportion of clusters containing fewer than four samples remains below 5.7%, indicating that batch-local clustering is stable and well-behaved. Moreover, for clusters containing only a few samples (<4), we draw negative examples from all other video and text prototypes within the same batch, ensuring that

Table 12: **Ablation study of PRN with and without category labels for clustering.**

| Method | R@1↑ | R@5↑ | R@10↑ | MdR↓ | MnR↓ | R@1↑ | R@5↑ | R@10↑ | MdR↓ | MnR↓ | SumR↑ |
|---|---|---|---|---|---|---|---|---|---|---|---|
| TempMe | 46.1 | 71.8 | 80.7 | - | 14.8 | 45.6 | 72.4 | 81.2 | - | 10.2 | 397.8 |
| +PRN w/o label | 46.6 | 72.6 | 81.4 | 2.0 | 14.5 | 46.9 | 73.7 | 83.1 | 2.0 | 10.1 | 404.3 (+6.5) |
| +PRN w/ label | 46.8 | 72.3 | 82.6 | 2.0 | 14.2 | 47.3 | 74.0 | 83.3 | 2.0 | 10.4 | 406.3 (+8.5) |

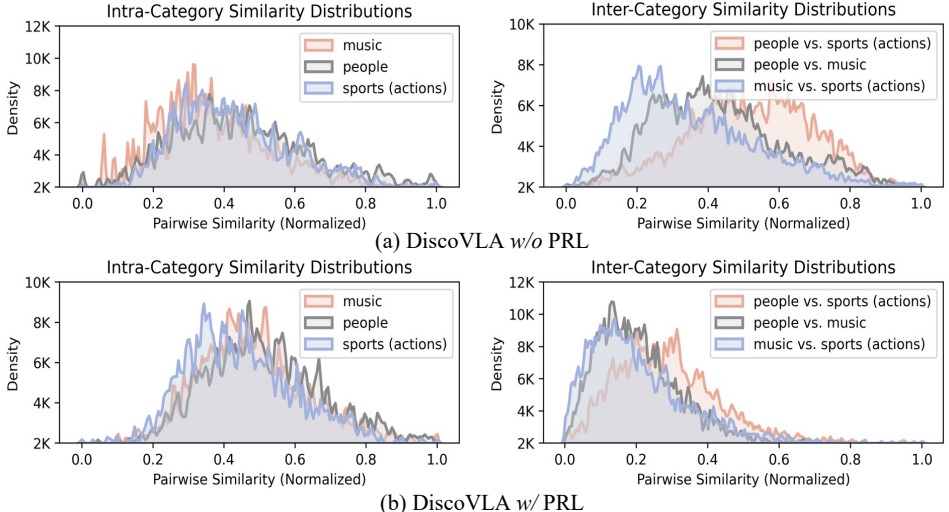

(a) DiscoVLA *w/o* PRL

(b) DiscoVLA *w/* PRL

Figure 14: **Intra- and Inter-category distance distributions of SOTA DiscoVLA *w/o* and *w/* our PRL on MSR-VTT dataset.**

the clusters remains well-posed and informative, even in the presence of underpopulated clusters, thereby maintaining stable and effective representation learning.

**Semantic Distributions for both Intra- and Inter-Category Samples.** To validate whether PRL effectively mitigates the IIC dilemma, we analyze the intra- and inter-category distance distributions on the MSR-VTT dataset. Specifically, we report the statistical distributions obtained from the SOTA DiscoVLA model with and without PRL in Fig. 14. We find that incorporating PRL leads to a noticeable increase in intra-category similarity while remaining within a suitable margin, indicating that PRL enhances feature compactness of the same category without causing collapse. Meanwhile, the inter-category similarity exhibits a clear decrease, demonstrating improved semantic separability across categories. These results collectively verify the effectiveness of PRL in alleviating the IIC dilemma and promoting a more discriminative representation space.

**Inter-modal and Intra-modal Overlap Analysis.** Inspired by (Kravets & Namboodiri, 2024), we test whether the learned features obtain good representations. We adopt CLIP4Clip and TempMe as baseline models and evaluate on MSR-VTT dataset. In detail, we first compute the similarities of instances within and between video clusters under two settings: *w/o* and *w/* PRL. Then we perform the same analysis for the text clusters. At last, we measure the cross-modal similarities between paired and unpaired video-text clusters. The results are shown in Fig. 15. We observe that PRL substantially reduces the overlap among clusters within each modality, indicating that the regularization loss helps create more compact and better-separated clusters in both the video and text feature spaces. In addition, for cross-modal prototypes, the overlap between matched and mismatched pairs is also greatly diminished, suggesting that PRL strengthens the alignment of true pairs while pushing apart semantically irrelevant ones. Together, these findings further demonstrate that PRL effectively shapes a more discriminative and well-structured representation space.

**Visualization of Retrieval Results.** We further provide qualitative text-to-video retrieval examples on ActivityNet in Fig. 16, using CLIP4Clip (Luo et al., 2022) as the baseline and comparing it with its PRL-enhanced counterpart. As shown in the examples, our method achieves strong retrieval performance in several cases, demonstrating its retrieval ability. However, PRL fails to bring successful results on some cases, as illustrated by the failure case where the retrieved video does not precisely match the text query. Possible reasons are: (1) we fix the hyperparameters of PRL for all datasets and methods. (2) ActivityNet often contains highly complex and diverse visual scenes that can weaken the strength of PRL's regularization. Even so, the retrieved results with PRL generally remain more semantically similar to the ground-truth videos than those of the baseline. These observations indicate that PRL consistently improves semantic alignment, even under challenging conditions.

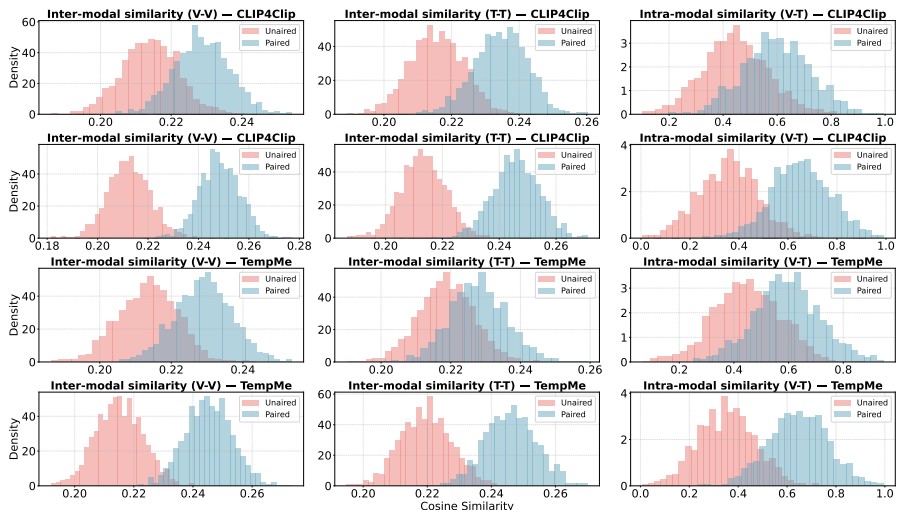

Figure 15: **Inter-modal and Intra-modal overlap measured as intersection area between cosine similarity distribution *w/o* PRL (first and third rows) and *w/* PRL (second and fourth rows).** First column: Inter-modal similarity of video modality. Second column: Inter-modal similarity of text modality. Third column: Intra-modal similarity between video and text modalities. We test two methods including base model CLIP4Clip and SOTA TempMe on MSR-VTT dataset.

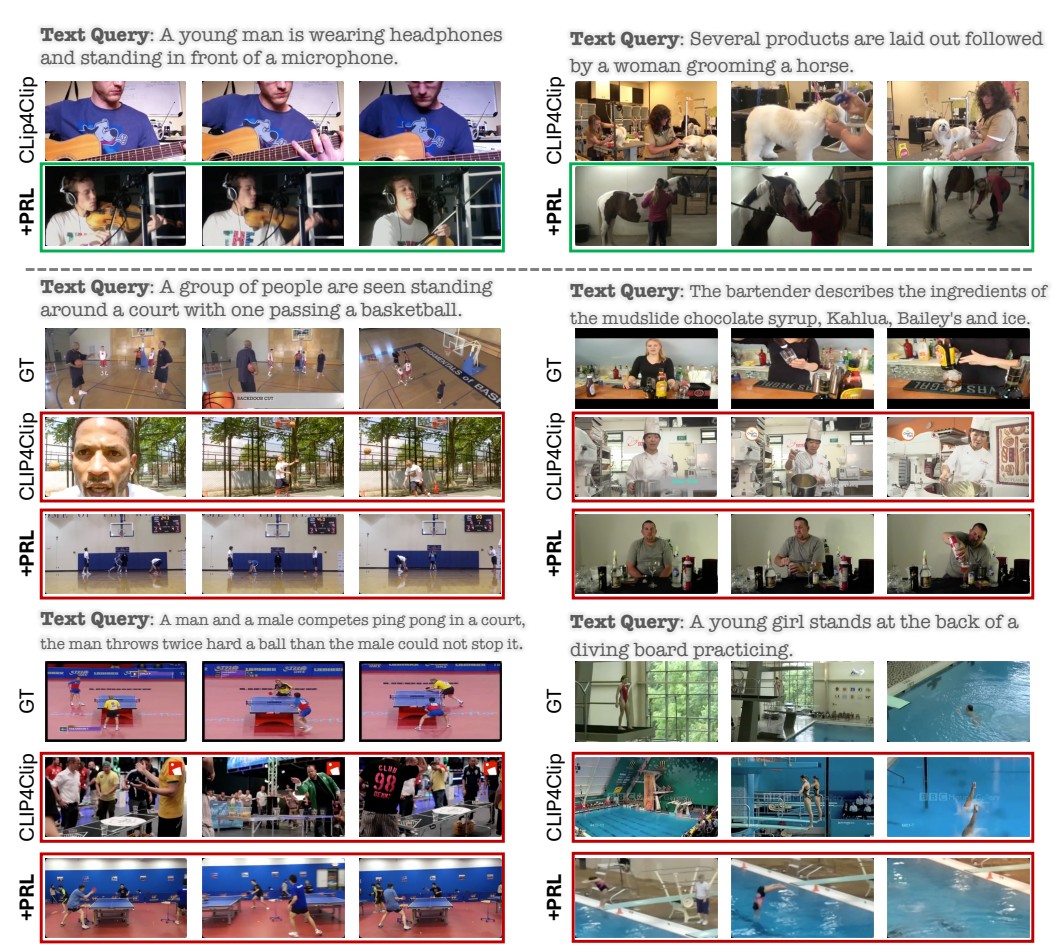

Figure 16: **Qualitative analysis of text-to-video results of CLIP4Clip (Luo et al., 2022) *w/o* or *w/* our PRL on ActivityNet dataset.** Given the text query, we provide the R@1 results. Correct and incorrect retrieved videos are highlighted in green and red, respectively.

