# OpenReview forum: "Prototype-based Regularization Learning For Text-Video Retrieval"
_ICLR.cc/2026/Conference — Submitted to ICLR 2026_

### Official Review · Reviewer_XRSn · 2025-10-26

**Soundness:** 3
**Presentation:** 3
**Contribution:** 2
**Rating:** 6
**Confidence:** 2

**Summary:**

This paper identifies a key problem in text-video retrieval called the Intra-Inter Conflict (IIC) dilemma: Intra-category variance: Videos/texts of the same category (e.g., "drinking") have inconsistent feature representations. Inter-category similarity: Videos/texts from different categories (e.g., "drinking" vs. "pouring") have entangled features.

To solve this, the authors propose Prototype-based Regularization Learning (PRL), a model-agnostic framework that uses clustering to find category-level "prototypes." It then uses two loss functions: (i) Prototype Discriminating Loss (PDL), which clusters similar instances around their prototype and pushes different prototypes apart. (ii) prototype Projection Loss (PPL), which aligns video and text prototypes in a shared semantic space.

**Strengths:**

**Clear Problem Defination**: This paper provides a clear analysis and nomenclature (IIC) for a widespread but often overlooked issue in retrieval models.

**Model-Agnostic and Flexible**: PRL is designed as a plug-and-play component that can be integrated into various existing models (e.g., CLIP4Clip, X-Pool) without major architectural changes.

**Effectiveness**: The method demonstrates significant and consistent performance improvements across five different datasets and multiple strong baseline models.

**Weaknesses:**

There are relatively few references, lacking some famous methods in this field.

I will consider increasing the score if other reviewers do not have so much concern.

**Questions:**

How does this work compare to vision-language-model for embedding such as VLM2Vec, LLaVE and UNITE? These VLM-based retrieval methods are good at semantic matching.

---

> ### Author Response · Authors · 2025-11-27
> **Response to Reviewer XRSn**
>
> We thank reviewer XRSn for the valuable time and constructive feedback. We provide point-to-point response below.
>
> > **W-1**: There are relatively few references, lacking some famous methods in this field.
>
> **Response:** We thank the reviewer for pointing out the issue. Since you did not provide references, we have made the following revisions to make the paper more comprehensive. In Sec. 4, we have provided a detailed discussion of other prototype-based methods, including PAN[1], ACP[2], and PAU[3] as suggested by reviewer F1QS. We further conduct direct comparisons between our approach and PAU on text-video retrieval and text-image retrieval tasks in Tab. 4 and Tab. 9. Moreover, we have also compared PRL with ACP on more challenging cross-domain text-video retrieval task in Tab.5. Please see **Response-W1-F1QS** for experimental results.
>
> ***Ref:***
>
> [1] Pan: Prototype-based adaptive network for robust cross-modal retrieval. SIGIR 2021.
>
> [2] Adaptive cross-modal prototypes for cross-domain visual-language retrieval. CVPR 2021.
>
> [3] Prototype-based aleatoric uncertainty quantification for cross-modal retrieval. NeurIPS 2023.
>
> > **Q-1**: How does this work compare to vision-language-model for embedding such as VLM2Vec, LLaVE and UNITE? These VLM-based retrieval methods are good at semantic matching.
>
> **Response:**  Thanks for your kind suggestion. We have conducted this experiment by integrating our PRL into the mainstream vision-language models including VLM2Vec, LLaVE and UNITE. As shown below, introducing PRL further improves the performance across most metrics on various tasks, strongly validating the robustness of the proposed method.
>
> | Method             | #Param | Classification | VQA      | Retrieval | Grounding | IND      | OOD      | Overall  |
> | ------------------ | ------ | -------------- | -------- | --------- | --------- | -------- | -------- | -------- |
> | # Datasets →       |        | 10             | 10       | 12        | 4         | 20       | 16       | 36       |
> | VLM2Vec (Qwen2-VL) | 7B     | 62.6           | 57.8     | 69.9      | 81.7      | 72.2     | 57.8     | 65.8     |
> | +PRL (Ours)        |        | **63.2**       | **57.9** | **70.0**  | **82.3**  | **72.8** | **58.5** | **66.1** |
> | LLaVE              | 0.5B   | 57.4           | 50.3     | 59.8      | 82.9      | 64.7     | 52.0     | 59.1     |
> | +PRL (Ours)        |        | **57.8**       | **50.7** | 59.7      | **83.2**  | **65.4** | 51.8     | **59.3** |
> | LLaVE              | 2B     | 62.1           | 60.2     | 65.2      | 84.9      | 69.4     | 59.8     | 65.2     |
> | +PRL (Ours)        |        | **62.9**       | **60.9** | **66.0**  | **85.4**  | **70.2** | **60.1** | **65.9** |
> | LLaVE              | 7B     | 65.7           | 65.4     | 70.9      | 91.9      | 75.0     | 64.4     | 70.3     |
> | +PRL (Ours)        |        | **66.2**       | **65.9** | **71.5**  | 91.8      | **75.6** | **64.8** | **70.7** |
> | Unite_instruct     | 2B     | 63.2           | 55.9     | 65.4      | 75.6      | 65.8     | 60.1     | 63.3     |
> | +PRL (Ours)        |        | **63.9**       | **56.5** | **66.1**  | **76.0**  | **65.4** | **60.7** | **63.9** |
> | Unite_instruct     | 7B     | 68.3           | 65.1     | 71.6      | 84.8      | 73.6     | 66.3     | 70.3     |
> | +PRL (Ours)        |        | **68.9**       | **65.7** | **72.2**  | **85.2**  | **74.2** | **66.9** | **70.9** |
>
> We have addde this experiment to the **Sec. 3.4, Tab.8** of the revised version.
>
> ***
> Finally, we would like to sincerely thank you for your careful review and valuable comments. We have revised our manuscript with discussions and experimental results according to your review. It is welcome if there are any further questions or suggestions.

---

### Official Review · Reviewer_QX4L · 2025-10-30

**Soundness:** 3
**Presentation:** 3
**Contribution:** 3
**Rating:** 4
**Confidence:** 4

**Summary:**

This paper aims to solve the Intra-Inter Conflict (IIC) dilemma in text-video retrieval. Through analysis, the authors find that the root cause of this problem is "appearance bias", where the model tends to focus on superficial semantics shared across samples while ignoring truly discriminative semantics.

To address this issue, the paper proposes a framework called "Prototype-based Regularization Learning" (PRL). PRL is a model-agnostic and orthogonal module that can be plugged-and-played into existing TVR models.

The core mechanisms of PRL include:

Clustering-based Prototype Mining, In each training mini-batch, K-means clustering is performed on video features to construct Video Prototypes, and paired text features are grouped into Text Prototypes.
1.  Prototype Discriminating Loss (PDL):** This loss function pushes different prototypes apart at the "Category-level" and encourages instances (hard negatives) within the same cluster to repel each other at the "Instance-level".
2.  Prototype Projection Loss (PPL): This loss function is used to address the ambiguity of text descriptions (e.g., "a person is walking"). It enhances cross-modal correspondence by pulling all text features within a cluster closer to their corresponding "cross-modal prototype".

Experimental results show that PRL improves the performance of six mainstream TVR baseline models (e.g., CLIP4Clip, X-Pool, TempMe) on five benchmark datasets, including MSR-VTT, MSVD, and DiDeMo. For example, on MSR-VTT, PRL brings a +6.5% and +5.0% SumR improvement to TempMe and CLIP4Clip, respectively. The method is also shown to be generalizable to image-text retrieval tasks.

**Strengths:**

1.  As a "plug-and-play" regularization module, PRL has high practical value. It does not propose a completely new, large-scale model, but rather regularizes the embedding space of existing models by mining prototypes, forcing the model to learn more compact and distinctive representations.
2.  High computational efficiency: PRL's biggest highlight is its efficiency. Experiments (Table 4) show that it introduces minimal training time (e.g., +0.59min) and memory overhead (e.g., +3MB) while adding 0 learnable parameters.

**Weaknesses:**

1.  The motivation of the paper is based on video retrieval, but the problems raised are widely present in clip-based retrieval tasks, which does not highlight the necessity of video retrieval as the main task.
2.  PRL's prototype mining is based on K-means performed on each mini-batch. If the data distribution within a mini-batch is biased, the quality of the mined prototypes may degrade. The paper does not discuss how to ensure clustering stability or analyze the impact of different Batch Sizes on prototype quality.
3.  The video prototypes are mined through *unsupervised* K-means clustering and do not use true category labels.

**Questions:**

1.  The paper claims that the model tends to focus on superficial semantics shared across samples. However, in the example in Figure 1 (bottom right), the top-5 retrieval results are all related to "sports," which aligns with the category. The unsupervised clustering operation does not seem to directly solve this issue. The authors need to provide statistical results with PRL , consistent with Figure 1 (top right).
2.  Some papers [1] state that CLIP focuses on cross-modal differences rather than intra-modal differences. Does this phenomenon lead to clustering instability? The authors need to statistically analyze the clustering results for visual and text modalities separately and compare their similarities and differences.
3.  CLIP exhibits  "bag-of-words" [2]. The reviewer is concerned that clustering might amplify this problem. Please provide an explanation.
4. Video retrieval pays more attention to the temporal problem, and the authors are asked to explain the necessity of using clustering methods in video tasks rather than image retrieval tasks.

[1] CLIP Adaptation by Intra-modal Overlap Reduction
[2] When and why vision-language models behave like bags-of-words, and what to do about it?

---

> ### Author Response · Authors · 2025-11-27
> **Response to Reviewer QX4L (1/4)**
>
> We thank reviewer QX4L for the valuable time and constructive feedback. We provide point-to-point response below.
>
> ***
>
> > **W-1**: The motivation of the paper is based on video retrieval, but the problems raised are widely present in clip-based retrieval tasks, which does not highlight the necessity of video retrieval as the main task.
>
> **Response:** Our apologies. We make detailed clarification below. In clip-based retrieval tasks, text-video retrieval involves more complex visual representations and offers greater practical value than others. Therefore, we treat it as our main task in the current version. We acknowledge that the IIC dilemma is a common issues in CLIP-based retrieval tasks. To this, we conduct experiments on text-image retrieval task to validate our motivation in Tab. 7. Moreover, we also testify RPL on more challenging multimodal embedding tasks in Tab. 8 as suggested by reviewer XRSn. PRL improves the current vision-language models across all metrics, showing its superiority.
>
> In response to your insightful suggestions, we will revise the paper title to _Prototype-based Regularization Learning for Cross-Modal Retrieval_ and update the narrative throughout the paper to more clearly position our method within the broader cross-modal retrieval setting. Given the scope of revision and the limited time available in the rebuttal phase, these refinements will be fully incorporated in the final version. We sincerely appreciate your constructive feedback, which has significantly improved the clarity and positioning of our work.

---

> ### Author Response · Authors · 2025-11-27
> **Response to Reviewer QX4L (2/4)**
>
> > **W-2**: PRL's prototype mining is based on K-means performed on each mini-batch. If the data distribution within a mini-batch is biased, the quality of the mined prototypes may degrade. The paper does not discuss how to ensure clustering stability or analyze the impact of different Batch Sizes on prototype quality.
>
> **Response:**  Mini-batch distribution bias primarily leads to two potential issues: empty clusters and assignment ties. We address empty clusters using a standard remedy—re-initializing them with a randomly selected sample from the same batch. For ties, we adopt a fixed rule that assigns the sample to the cluster with the smallest centroid index, consistent with common K-means implementations and avoiding ambiguity (**Line 200-203**).
>
> Further, we have analyzed clustering stability in Fig. 13 of Appendix. In detail, we evaluate the distribution of samples within each cluster across all batches during trainin on four representative models  (CLIP4Clip, X-CLIP, DiscoVLA, and TempMe) across five datasets  (MSVD, MSR-VTT, DiDeMo, LSMDC, and ActivityNet). We observe that the proportion of clusters containing fewer than four samples is consistently below 5.7%, suggesting that the batch-local clustering process remains stable and well-behaved in practice. For the very few small clusters that do occur, we draw negative samples from all remaining video/text prototypes within the same batch, ensuring that the contrastive objective remains well-posed and that the learning signal is not degraded by insufficient intra-cluster negatives.
>
> This discussion and experiment are incorporated into the **Appendix Sec.C and Fig. 13**.
>
> Finally, we take Temp as baseline model and examine the effect of different batch sizes for clustering. Due to limitations in available computational resources, we constrain the maximum batch size to 512. As observed below, PRL consistently improves the baseline method across all batch size settings. The improvement becomes more pronounced with larger batch sizes, likely because more samples provide richer semantic information, enhancing the discriminability of clusters and prototypes, and thereby strengthening the regularization effect.
>
> | Method    | R@1↑     | R@5↑     | R@10↑    | MdR↓ | MnR↓     | R@1↑     | R@5↑     | R@10↑    | MdR↓ | MnR↓     | SumR↑            |
> | --------- | -------- | -------- | -------- | ---- | -------- | -------- | -------- | -------- | ---- | -------- | ---------------- |
> | B=64  |          |          |          |      |          |          |          |          |      |          |                  |
> | TempMe    | 45.1     | 71.0     | 80.0     | 2.0  | 15.8     | 45.4     | 72.0     | 81.5     | 2.0  | 11.7     | 395.0            |
> | **+PRL**  | **46.3** | **72.2** | **81.1** | 2.0  | **14.8** | **46.0** | **72.8** | **82.6** | 2.0  | **10.6** | **402.0** (+6.0) |
> | B=128 |          |          |          |      |          |          |          |          |      |          |                  |
> | TempMe    | 46.1     | 71.8     | 80.7     | -    | 14.8     | 45.6     | 72.4     | 81.2     | -    | 10.2     | 397.8            |
> | **+PRL**  | **46.6** | **72.6** | **81.4** | 2.0  | **14.5** | **46.9** | **73.7** | **83.1** | 2.0  | **10.1** | **404.3** (+6.5) |
> | B=256 |          |          |          |      |          |          |          |          |      |          |                  |
> | TempMe    | 46.7     | 72.5     | 81.2     | 2.0  | 14.5     | 46.8     | 73.5     | 82.0     | 2.0  | 10.0     | 402.7            |
> | **+PRL**  | **46.9** | **73.4** | **82.8** | 2.0  | **14.3** | **47.4** | **74.6** | **84.3** | 2.0  | **9.9**  | **409.4** (+6.7) |
> | B=512 |          |          |          |      |          |          |          |          |      |          |                  |
> | TempMe    | 47.0     | 72.7     | 81.4     | 2.0  | 14.5     | 47.2     | 73.8     | 83.0     | 2.0  | 9.8      | 405.1            |
> | **+PRL**  | **47.5** | **73.9** | **82.6** | 2.0  | **14.4** | **48.7** | **74.4** | **85.0** | 2.0  | **9.6**  | **412.1** (+7.0) |
>
> This discussion and experiment have been added into the **Appendix Sec.B and Tab. 11.**

---

> ### Author Response · Authors · 2025-11-27
> **Response to Reviewer QX4L (3/4)**
>
> > **W-3**: The video prototypes are mined through *unsupervised* K-means clustering and do not use true category labels.
>
> **Response:**  Considering this, we conduct the experiment by use true category label for K-means clustering on MSR-VTT. Indeed, this strategy results in slightly better performance than *unsupervised* manner as shown below:
>
> | Method         | R@1  | R@5  | R@10 | MdR  | MnR  | R@1  | R@5  | R@10 | MdR  | MnR  | SumR          |
> | -------------- | ---- | ---- | ---- | ---- | ---- | ---- | ---- | ---- | ---- | ---- | ------------- |
> | CLIP4Clip      | 46.1 | 71.8 | 80.7 | -    | 14.8 | 45.6 | 72.4 | 81.2 | -    | 10.2 | 397.8         |
> | +PRN w/o label | 46.6 | 72.6 | 81.4 | 2.0  | 14.5 | 46.9 | 73.7 | 83.1 | 2.0  | 10.1 | 404.3 (+6.5)  |
> | +PRN w/ label  | 46.8 | 72.3 | 82.6 | 2.0  | 14.2 | 47.3 | 74.0 | 83.3 | 2.0  | 10.4 | 406.3 (+8.5） |
>
> However, the reasons of we use *unsupervised* K-means clustering are follows:
>
> **First**, unsupervised prototypes capture invariant visual/text patterns that exist across categories, enabling better transferability to unseen classes and reducing dependence on potentially noisy or biased annotations.
>
> **Second**, it is hard to define a clear category boundary since videos usually contain multiple concepts. Using unsupervised clustering at the feature level avoids this ambiguity and lets the model automatically group samples by their real semantic patterns.
>
> **Third**, label-based clustering strategy relies on datasets with heavy category annotations, which are often expensive and labor-intensive to acquire.
>
> To this, we adopt unsupervised K-means clustering in RPL.
>
> This experiment and discussion is added in **Appendix Sec.B and Table 12.** Thanks.
>
> ***
>
> > **Q-1**: The paper claims that the model tends to focus on superficial semantics shared across samples. However, in the example in Figure 1 (bottom right), the top-5 retrieval results are all related to "sports," which aligns with the category. The unsupervised clustering operation does not seem to directly solve this issue. The authors need to provide statistical results with PRL , consistent with Figure 1 (top right).
>
> **Response:**  Although the retrieved videos in Fig. 1 (bottom right) all fall under the broad “sports’’ category, none of them correspond to the key action semantics in the query, such as *“play tennis’’*. Instead, the model primarily attends to superficial cues, *e.g.*, generic sports scenes with similar visual backgrounds, the common phrase "two men", "a huge crowd", while failing to capture the fine-grained and discriminative semantics expressed in the text. This indicates that superficial appearance similarity, rather than true semantic alignment, dominates the retrieval process even when samples share the same category.
>
> This problem is considered within each clusters, *i.e.*, $\mathcal{L} _ {PDL_C}$, treats intra-cluster samples as hard negatives, which maintains compact representations while learns fine-grained separation between each other.
>
> As suggested, we have provided the statistical results with PRL which consistent with Figure 1 (top right) in **Appendix Sec. C with Fig.14**. We observe that adding PRL increases intra-category similarity while keeping it within a reasonable margin, meaning PRL makes features of the same category more compact without collapsing them. At the same time, inter-category similarity clearly decreases, showing stronger separation between different semantics. Together, these results confirm that PRL effectively alleviates the IIC dilemma and leads to a more discriminative feature space.
>
> > **Q-2**: Some papers [1] state that CLIP focuses on cross-modal differences rather than intra-modal differences. Does this phenomenon lead to clustering instability? The authors need to statistically analyze the clustering results for visual and text modalities separately and compare their similarities and differences.
>
> **Response:** We have conduced this analyze statistically in the revised version. We find that most of prototypes receives stable instance assignments as shown in **Appendix Fig. 13**. The clustering instabilities will causes two main problems: empty clusters or tie cases, we discusse this in the response of **W-2**.
>
> Following **[1]**, We evaluate whether PRL improves feature representations using CLIP4Clip and TempMe on MSR-VTT dataset. We compute intra- and inter-cluster similarities for videos and texts, as well as cross-modal similarities between paired and unpaired video-text clusters, with and without PRL. The experiment are shown in **Fig. 15 of Appendix**. As observed, PRL substantially reduces intra- and inter-cluster overlap within each modality and decreases cross-modal overlap between mismatched pairs.

---

> ### Author Response · Authors · 2025-11-27
> **Response to Reviewer QX4L (4/4)**
>
> > **Q-3**: CLIP exhibits "bag-of-words" [2]. The reviewer is concerned that clustering might amplify this problem. Please provide an explanation.
>
> **Response:**  We appreciate the reviewer’s concern about clustering potentially amplifying CLIP’s bag-of-words (BoW) issue, but our strategy **mitigates rather than exacerbates** this problem for three core reasons:
>
> 1. Our clustering is based on holistic video and text embeddings, which already encode temporal and contextual cues. Thus, the clustering step is able to poentially group samples that share lexical elements but differ in compositional logic (e.g., "black coat + blue sky" vs. "blue coat + black sky") into the same prototype.
> 2. Prototype learning encourages similar video segments to be grouped and contrasted at a coarse semantic level. This aggregation explicitly forces separation between semantically distinct events, which counteracts the low-level BoW tendency of mixing unrelated instances.
> 3. Intra-cluster contrastive learning forces the model to encode fine-grained compositional details (attribute-object binding, spatial relations) instead of relying on shortcut single words to distinguish samples, i.e., hard negative learning within each cluster, which is strongly consistent with the idea mentioned in Sec. 4 (2) of the [2] : “*2. Sampling strong alternative images: To generate... we first use CLIP to compute the pairwise similarity between all images in the dataset. During training, for each image in the batch, we sample one of the K = 3 nearest neighbors as the strong alternative image*...”
>
> ***
>
> > **Q-4**: Video retrieval pays more attention to the temporal problem, and the authors are asked to explain the necessity of using clustering methods in video tasks rather than image retrieval tasks.
>
> **Response**:  We acknowledge that temporal modeling is indeed an important aspect in video retrieval. However, the primary goal of our work is not to investigate temporal dynamics, but rather to explore more discriminative features at the representative level. Many existing studies have also focused on improving representation learning and leave temporal structures unchanged[1,2,3].
>
> It is important to emphasize that our method is a plug-and-play design for cross-modal retrieval. We have indeed conducted generalization experiments on image retrieval tasks. Moreover, following reviewer XRSn’s suggestion, we further validated the effectiveness of our approach on vision-language–based embedding tasks in Tab. 8.
>
> As discussed in the response of W-1, we will revise the title and content accordingly to better reflect that our method is a general cross-modal retrieval framework.
>
> **Ref:**
>
> [1] Expectation-Maximization Contrastive Learning for Compact Video-and-Language Representations. NeurIPS 2022.
>
> [2] Dual Alignment Unsupervised Domain Adaptation for Video-Text Retrieval. CVPR 2023.
>
> [3] Rebalancing Contrastive Alignment with Bottlenecked Semantic Increments in Text-Video Retrieval. NeurIPS 2025.
>
> ***
>
> Finally, we would like to sincerely thank you for your careful review and valuable comments. We have revised our manuscript with discussions and experimental results according to your review. It is welcome if there are any further questions or suggestions.

---

### Official Review · Reviewer_F1QS · 2025-10-31

**Soundness:** 3
**Presentation:** 3
**Contribution:** 2
**Rating:** 4
**Confidence:** 4

**Summary:**

This paper presents Prototype-based Regularization Learning (PRL) for text-video retrieval. The method addresses the Intra–Inter Conflict (IIC) dilemma—how to maintain compactness within categories while ensuring clear separability between categories in joint video–text embedding spaces.
PRL introduces a clustering-based prototype mining approach with two new regularization losses: 1) Prototype Discriminating Loss (PDL) for semantic separation, and 2) Prototype Projection Loss (PPL) for adaptive cross-modal alignment.

The framework is model-agnostic and can be easily integrated into existing retrieval models. Experiments across five major datasets and multiple architectures show consistent performance gains. The paper also includes ablation studies and visualizations to support its claims.

**Strengths:**

1.  The paper gives a strong motivation for the IIC dilemma, supported by both visual and analytical explanations (see Figure 1, Page 2). The empirical analysis convincingly demonstrates drift and entanglement issues in current approaches.

2.  PRL is plug-and-play, requiring minimal architectural changes and maintaining efficiency.

**Weaknesses:**

1.  The paper does not sufficiently compare PRL with recent prototype-based or IIC-focused methods. Several closely related works [a-c] are absent from both the discussion and experiments. This omission weakens the paper’s claim of novelty and superiority.

[a]  Zeng Z, Wang S, Xu N, et al. Pan: Prototype-based adaptive network for robust cross-modal retrieval[C]//Proceedings of the 44th international ACM SIGIR conference on research and development in information retrieval. 2021: 1125-1134.

[b]  Liu Y, Chen Q, Albanie S. Adaptive cross-modal prototypes for cross-domain visual-language retrieval[C]//Proceedings of the IEEE/CVF conference on computer vision and pattern recognition. 2021: 14954-14964.

[c]  Li H, Song J, Gao L, et al. Prototype-based aleatoric uncertainty quantification for cross-modal retrieval[J]. Advances in Neural Information Processing Systems, 2023, 36: 24564-24585.

2.  The clustering and assignment process in Section 2.2 is underspecified. It is unclear how prototypes and cluster memberships are maintained across batches or epochs, and how issues like empty clusters or ties are handled. Some loss definitions (e.g., Eq. 5) lack clarity on normalization and contrastive sampling, especially for small clusters.

**Questions:**

1. How does batch-wise K-means handle non-uniform batch composition? Are prototypes re-initialized each batch or tracked across epochs?

2. Have you tried soft or probabilistic clustering (e.g., GMM)? Why exclusively use hard K-means?

---

> ### Author Response · Authors · 2025-11-27
> **Response to Reviewer F1QS (1/3)**
>
> We thank reviewer F1QS for the valuable time and constructive feedback. We provide point-to-point response below.
>
> ***
>
> > **W-1**: The paper does not sufficiently compare PRL with recent prototype-based or IIC-focused methods. Several closely related works [a-c] are absent from both the discussion and experiments. This omission weakens the paper’s claim of novelty and superiority.
>
> **Response:** Thank you for pointing this out. We have added a detailed discussion of these works [a,b,c] in the **Sec. 4-Prototype Learning** of the revised version:
>
> **Line 520-522**: (Zeng et al., 2021) introduces PAN, a prototype-based method that tackles unknown-category queries and modality imbalance to enhance the robustness of cross-modal retrieval.
>
> **Line 522-524:** (Liu et al., 2021) proposes Adaptive Cross-Modal Prototypes, a method that enables cross-domain visual–text retrieval with unlabeled target data by preserving compositional structure and reducing distribution shift.
>
> **Line 524-526**: (Li et al., 2023a) devises PAU that quantifies aleatoric uncertainty using evidential theory to provide more reliable cross-modal retrieval under low-quality or ambiguous data.
>
> As suggested, we conduct experiments as follows. Since [a] does not provide open-source code and its target tasks differ from ours, we only discuss it as above. We have compared PRL with [c] on both text-video retrieval and text-image retrieval tasks, and further compared with [b] on more challenging cross-domain text-video retrieval. The results are shown below:
>
> | MSR-VTT        |          |          | t2v      |      |          |          |          | **v2t**  |      |         |                   |
> | -------------- | -------- | -------- | -------- | ---- | -------- | -------- | -------- | -------- | ---- | ------- | ----------------- |
> | Method         | R@1↑     | R@5↑     | R@10↑    | MdR↓ | MnR↓     | R@1↑     | R@5↑     | R@10↑    | MdR↓ | MnR↓    | SumR↑             |
> | Baseline [c]   | 45.6     | 72.9     | 81.5     | 2.0  | 14.5     | 45.2     | 71.6     | 81.5     | 2.0  | 10.9    | 398.3             |
> | PAU [c]        | 48.5     | 72.7     | **82.5** | 2.0  | 14.0     | 48.3     | **73.0** | 83.2     | 2.0  | 9.7     | 408.2 (+9.9)      |
> | **PRL (ours)** | **48.9** | **73.0** | **82.5** | 2.0  | **13.8** | **48.5** | **73.0** | **83.6** | 2.0  | **9.4** | **409.4** (+11.1) |
> ***
> |                |  |          |     MSCOCO 1K     |          |          |          |  |          |     MSCOCO 5K     |          |          |          |
> | :------------- | --------- | -------- | -------- | -------- | -------- | -------- | --------- | -------- | -------- | -------- | -------- | -------- |
> |                |           | i2t      |          |          | t2i      |          |           | i2t      |          |          | t2i      |          |
> | Method         | R@1↑      | R@5↑     | R@10↑    | R@1↑     | R@5↑     | R@10↑    | R@1↑      | R@5↑     | R@10↑    | R@1↑     | R@5↑     | R@10↑    |
> | Baseline [c]   | 80.1      | 95.7     | 98.2     | 67.1     | 91.4     | 96.6     | 62.9      | 84.9     | 91.6     | 46.5     | 73.8     | 82.9     |
> | PAU [c]        | 80.4      | 96.2     | 98.5     | 67.7     | 91.8     | 96.6     | 63.6      | 85.2     | 92.2     | 46.8     | 74.4     | 83.7     |
> | **PRL (ours)** | **80.9**  | **96.7** | **98.6** | **67.8** | **92.0** | **96.9** | **63.8**  | **85.5** | **92.4** | **46.9** | **74.4** | **84.0** |
>
> ***
>
> | Method         | MSR-VTT  | →  | MSVD  |          |          |       | MSVD    | →        | MSR-VTT |         |          |         |
> | -------------- | -------- | -------- | ----- | -------- | -------- | ----- | ------- | -------- | ------- | ------- | -------- | ------- |
> |                | t2i      |          |       | i2t      |          |       | t2i     |          |         | i2t     |          |         |
> |                | R@1↑     | R@10↑    | MnR↓  | R@1↑     | R@10↑    | MnR↓  | R@1↑    | R@10↑    | MnR↓    | R@1↑    | R@10↑    | MnR↓    |
> | Baseline       | 14.2     | 52.3     | 9     | 16.6     | 50.0     | 10    | 3.6     | 17.2     | 98      | 2.5     | 13.5     | 117     |
> | ACP [b]        | 16.6     | 55.2     | 8     | 22.1     | 52.5     | 8     | 4.4     | 17.9     | 97      | 3.1     | 15.3     | 111     |
> | **PRL (Ours)** | **17.8** | **56.9** | **7** | **23.4** | **53.4** | **7** | **6.5** | **18.5** | **92**  | **4.9** | **16.8** | **104** |
>
> As observed, our method consistently outperform the recent prototype-based methods, showing its superiority.
>
> We have added these experiments in the revised version, presented in **Sec. 3.3 with Tab.4 and Tab. 5** of the main paper, and **Appendix Sec. B with Tab. 9** due to page limitation.

---

> ### Author Response · Authors · 2025-11-27
> **Response to Reviewer F1QS (2/3)**
>
> > **W-2**: The clustering and assignment process in Section 2.2 is underspecified. It is unclear how prototypes and cluster memberships are maintained across batches or epochs, and how issues like empty clusters or ties are handled. Some loss definitions (e.g., Eq. 5) lack clarity on normalization and contrastive sampling, especially for small clusters.
>
> **Response**: For each mini-batch $\mathcal{B}$, we perform independent K-means clustering on the video/text embeddings  $\mathcal{Z}^v = \\{z_i^v\\}_{i=1}^B$, $\mathcal{Z}^t = \\{z _ i^t\\} _ {i=1}^B$. The clustering is recomputed from scratch for every batch and epochs.  The Prototypes $\mathcal{P}^v$, text prototypes $\mathcal{P}^t$, and cross-modal prototypes $\mathcal{P}^c$  belong only to that batch. Thus, prototypes are batch-dependent and decoupled across training iterations, preventing drift or accumulation of clustering errors.
>
> To better quantify the clustering assignments, we quantified the sample distribution of each batch cluster in **Appendix, Fig. 13**. As observed,  most clusters achieved stable sample assignments, with most 5.7% of clusters having fewer than 4 samples. To make the clustering more stable, we follow a standard remedy: Empty clusters are re-initialized using a randomly selected sample from the current batch. Since clustering is batch-local, this simple strategy is stable and effective. For ties in cluster assignment, we adopt a fixed rule that assigns the sample to the cluster with the smallest centroid index. This matches the tie-breaking strategy used in common K-means implementations and avoids ambiguity.
>
> All contrastive losses apply a softmax normalization over the available positives and negatives within the current batch. To make the Eq. 5 clearer, we rewrite it as follows.
>
> $
> \mathcal{L}_{\\text{PPL}} = - \frac{1}{\\sum _ m |C _ m^t|}
> \sum _ {m=1}^{M} \sum _ {i \\in C _ m^t} \\log \\frac{
>     \\exp(\\langle z^t _ i, p^{c} _ m \\rangle / \\tau)
> }{
>     \\exp(\\langle z^t _ i, p^{c} _ m \\rangle / \\tau)
>     +
>         \\sum _ {\\substack{k=1 \\ k \ne m}}^{M}
>         \\exp(\\langle z^t_i, p^{c} _ k \rangle / \tau)
> }.
> $
>
> For each text feature $z^t_i$ belonging to cluster $C_m^t$, the pair $(z^t_i, p^c_m)$ forms the positive sample, while all other prototypes $\{p^c_k \mid k \ne m\}$ serve as negative samples. Moreover, if a cluster contains only very few samples, *_e.g._*, ($|C_m^t| = 1$ or 2), negatives are taken from other clusters within the same batch to keep the contrastive objective well-posed.
>
> We have updated **Sec. 2.2 and Eq. 5** to make it clear and easy to read in the revised version.
>
> ***
>
> > **Q-1**: How does batch-wise K-means handle non-uniform batch composition? Are prototypes re-initialized each batch or tracked across epochs?
>
> **Response**:  Since clustering is independent for each mini-batch, prototypes are re-initialized from scratch rather than tracked across epochs. This makes each batch forming its own local semantic prototypes, which naturally handles non-uniform batch composition.
>
> Additionally, we conduct experiment by tracking prototypes across epochs. We save the prototype features of each batch in the previous epoch and averages them, which serves as the initial prototype for the next epoch. The results are demonstrated below. We find that tracks prototypes across epochs degrades performance. Possible reason is that tracking prototypes across epochs tends to preserve outdated or noisy cluster centers. As the feature space changes every epoch, these stale prototypes no longer match the current representations and gradually drift away from meaningful semantics.
>
> | Method                           | R@1↑     | R@5↑     | R@10↑    | MdR↓ | MnR↓     | R@1↑     | R@5↑     | R@10↑    | MdR↓ | MnR↓     |
> | -------------------------------- | -------- | -------- | -------- | ---- | -------- | -------- | -------- | -------- | ---- | -------- |
> | CLIP4Clip                        | 42.8     | 70.0     | 79.8     | 2.0  | 16.5     | 42.0     | **70.9** | 79.7     | 2.0  | 12.0     |
> | +PRL (re-initialized each batch) | **43.1** | **70.9** | **80.6** | 2.0  | **16.1** | **43.4** | 70.7     | **81.5** | 2.0  | **11.6** |
> | +PRL (tracked across epochs)     | 42.9     | 70.4     | 80.2     | 2.0  | 16.3     | 43.1     | 70.5     | 81.1     | 2.0  | 11.8     |
> | TempMe                           | 46.1     | 71.8     | 80.7     | -    | 14.8     | 45.6     | 72.4     | 81.2     | -    | 10.2     |
> | +PRL (re-initialized each batch) | **46.6** | **72.6** | **81.4** | 2.0  | **14.5** | **46.9** | **73.7** | **83.1** | 2.0  | **10.1** |
> | +PRL (tracked across epochs)     | 46.3     | 72.1     | 81.0     | 2.0  | 14.7     | 46.5     | 73.0     | 82.6     | 2.0  | 10.4     |

---

> ### Author Response · Authors · 2025-11-27
> **Response to Reviewer F1QS (3/3)**
>
> > **Q-2**: Have you tried soft or probabilistic clustering (e.g., GMM)? Why exclusively use hard K-means?
>
> **Response:** According to your suggestion, we have replaced K-means with other clustering algorithms including GMM and DBSCAN, and the results are shown below. As observed, the performance variation is minor. Since the focus of this work is not on exploring clustering methods, we opted for K-means since it is efficient, easy to implement.
>
> | Method         | R@1↑     | R@5↑     | R@10↑    | MdR↓ | MnR↓     | R@1↑     | R@5↑     | R@10↑    | MdR↓ | MnR↓     |
> | -------------- | -------- | -------- | -------- | ---- | -------- | -------- | -------- | -------- | ---- | -------- |
> | **CLIP4Clip**      | 42.8     | 70.0     | 79.8     | 2.0  | 16.5     | 42.0     | 70.9     | 79.7     | 2.0  | 12.0     |
> | +PRL (K-means) | **43.1** | **70.9** | **80.6** | 2.0  | **16.1** | **43.4** | 70.7     | 81.5     | 2.0  | 11.6     |
> | +PRL (GMM)     | 42.9     | 69.7     | 80.1     | 2.0  | 16.5     | 42.9     | 70.8     | **81.6** | 2.0  | 12.3     |
> | +PRL (DBSCAN)  | 43.0     | 70.2     | 80.4     | 2.0  | 16.3     | 43.2     | **71.0** | 81.0     | 2.0  | **11.3** |
> | **DiscoVLA**       | 47.0     | 73.0     | **82.8** | -    | 14.1     | 47.7     | 73.6     | 83.6     | -    | 10.0     |
> | +PRL (K-means) | **47.4** | 73.1     | 82.4     | 2.0  | **14.0** | **48.0** | **73.8** | **83.7** | 2.0  | **9.3**  |
> | +PRL (GMM)     | 47.2     | **73.2** | 82.5     | 2.0  | **14.0** | 47.3     | **73.8** | 83.5     | 2.0  | 9.9      |
> | +PRL (DBSCAN)  | **47.4** | 73.1     | 82.6     | 2.0  | 14.1     | 47.5     | 73.5     | 83.4     | 2.0  | 10.0     |
> | **TempMe**         | 46.1     | 71.8     | 80.7     | -    | 14.8     | 45.6     | 72.4     | 81.2     | -    | 10.2     |
> | +PRL (K-means) | **46.6** | 72.6     | 81.4     | 2.0  | **14.5** | **46.9** | 73.7     | **83.1** | 2.0  | **10.1** |
> | +PRL (GMM)     | 46.1     | 72.3     | 81.7     | 2.0  | 14.7     | 45.7     | **74.0** | 82.0     | 2.0  | 10.2     |
> | +PRL (DBSCAN)  | 46.3     | 72.5     | 81.5     | 2.0  | 14.6     | 46.0     | 73.5     | 82.5     | 2.0  | 10.2     |
>
> We have added this experiments in the revised version in **Sec. B with Tab. 10 of Appendix**.
>
> ***
>
> Finally, we would like to sincerely thank you for your careful review and valuable comments. We have revised our manuscript with discussions and experimental results according to your review. It is welcome if there are any further questions or suggestions.

---

### Official Review · Reviewer_CTri · 2025-11-01

**Soundness:** 3
**Presentation:** 3
**Contribution:** 3
**Rating:** 4
**Confidence:** 5

**Summary:**

This work studies the task of text-video retrieval with a new method called prototype-based regularization learning. The proposed method is motivated by a new point termed as intra-inter conflict dilemma, where the category-consistent instances that display substantial distributional disparity and the  instances belonging to different categories exhibit distributional coupling. The proposed method is applied on a couple of methods and see some performance boost.

**Strengths:**

* This work study the complex representation space of the multimodal data and aims to deliver more discriminative representations for retrieval, which is encouraging.

* The proposed method is applied on a number of methods and extend to text-image retrieval, which show some generalization ability.

* The introduction and method are well-written.

**Weaknesses:**

* Despite the new concept of the Intra-Inter Conflict (IIC) dilemma is interesting, there are some concerns on it. It seems that the proposed dilemma lacks generalizability. Fig.1 shows the observation on MSRVTT and upon DisCoVLA. It is unclear if this dilemma is specific to the DiscoVLA or MSRVTT. More empirical evidence from other datasets or preliminary methods is needed to justify the proposed claim.

* The performance boost of the proposed method is generally marginal, doubting the necessity of the proposed challenge and the effectiveness of the proposed method.

* This work suggests that the semantics from the same category should be  more similar, while the ones from different categories should be less similar. I'm not sure whether this assumption always holds considering the complex behavior of the representation space.

* Besides, I'm not sure whether this assumption is always consistent with the contrastive learning objective. More empirical evidence and justifications are required to admit this intuition.

**Questions:**

* How the category in the proposed motivation is defined? To what extent some concepts can be regarded as the same category, for example, whether two dogs with different breeds belong to the same category? Or whether the two verbs that describe the same action belong the same category? Since this lay the foundation for the motivation and the proposed method, a more rigorous (or quantitative) definition of it is expected beyond the conceptual illustration.

* On some dataset, such as ActivityNet, the proposed method does not benefit but fails with CLIP3Clip, a failing case analyzation would be helpful.

* How to determine the number of the prototypes across different methods and datasets? Why more count does not lead to more fine-grained representation learning and consistent improvement?

---

> ### Author Response · Authors · 2025-11-27
> **Response to Reviewer CTri (1/3)**
>
> We thank reviewer CTri for the valuable time and constructive feedback. We provide point-to-point response below.
>
> ***
>
> > **W-1:** Despite the new concept of the Intra-Inter Conflict (IIC) dilemma is interesting, there are some concerns on it. It seems that the proposed dilemma lacks generalizability. Fig.1 shows the observation on MSRVTT and upon DisCoVLA. It is unclear if this dilemma is specific to the DiscoVLA or MSRVTT. More empirical evidence from other datasets or preliminary methods is needed to justify the proposed claim.
>
> **Response:** We further substantiate our claim by providing comprehensive empirical evidence across both text–video and text–image retrieval tasks. For text–video retrieval, we evaluate four representative and widely adopted approaches: CLIP4Clip, X-Pool, DisCoVLA, and TempMe, on five standard benchmarks (MSVD, MSR-VTT, DiDeMo, LSMDC, and ActivityNet). For text–image retrieval, we validate the presence of the IIC dilemma on Flickr30K and MSCOCO using two models, CLIP and BiCro. The complete analysis is provided in **Appendix Sec. A with Fig. 8 and Fig. 10**.
>
> Across all models and datasets, we consistently observe IIC dilemma, *i.e.*, category-consistent instances exhibit clear distributional inconsistency while category-divergent instances show unexpected cross-category entanglement — a common behavior that recurs regardless of the architecture or dataset. Such consistency strongly supports our argument that the proposed IIC dilemma represents a general and systematic phenomenon in current retrieval frameworks, rather than an artifact of particular methods or data.
>
> ***
>
> > **W-2:** The performance boost of the proposed method is generally marginal, doubting the necessity of the proposed challenge and the effectiveness of the proposed method.
>
> **Response**:  In text–video retrieval, annual SOTA gains on the key R@1 metric typically range from only 0.3% to 2.5% on MSR-VTT (2022–2025)[1,2,3,4]. As a plug-and-play module, PRL consistently yields non-trivial improvements across diverse TVR models, such as +0.5%, +1.6%, and +0.9% R@1 over TempMe (2025), DiCoSA (2023), and X-CLIP (2022), respectively. Beyond TVR, PRL also generalizes well to text–image retrieval. As shown in Tab. 7, applying PRL to models on Flickr30K and MSCOCO leads to consistent improvements across all evaluation metrics. Moreover, the additional experiment suggested by Reviewer XRSn (Tab. 8) further validates that PRL remains effective on LLM-based vision–language models for various multimodal embedding tasks.
>
> In parallel, PRL introduces only negligible computational overhead (Tab. 6), ensuring that its benefits do not come at the cost of increased model complexity or runtime.
>
> ***Ref:***
>
> [1] X-Pool: Cross-Modal Language-Video Attention for Text-Video Retrieval. CVPR 2022.
>
> [2] TS2-Net: Token Shift and Selection Transformer for Text-Video Retrieval. ECCV 2022.
>
> [3] Text-Video Retrieval with Disentangled Conceptualization and Set-to-Set Alignment. IJCAI 2023.
>
> [4] DGL: Dynamic Global-Local Prompt Tuning for Text-Video Retrieval. AAAI 2024.
>
> [5] TempMe: Video Temporal Token Merging for Efficient Text-Video Retrieval. ICLR 2025.
>
> ***
>
> > **W-3**: This work suggests that the semantics from the same category should be more similar, while the ones from different categories should be less similar. I'm not sure whether this assumption always holds considering the complex behavior of the representation space.
>
> **Response:** Thanks. We provide additional feature-space visualizations in **Appendix Sec. C (Fig. 9 and Fig. 11)**, covering six representative models across both text–video and text–image tasks. As shown, simply adding PRL, without modifying the training architecture or tuning any hyperparameters, consistently yields more separable and discriminative feature distributions compared to the counterparts without PRL (Fig. 8 and Fig. 10). These results offer direct and intuitive evidence that PRL improves representation quality and supports our core intuition.

---

> ### Author Response · Authors · 2025-11-27
> **Response to Reviewer CTri (2/3)**
>
> > **W-4**: Besides, I'm not sure whether this assumption is always consistent with the contrastive learning objective. More empirical evidence and justifications are required to admit this intuition.
>
> **Response:** Thanks. We would like to provide further clarification. Learning clear and discriminative representations in the embedding space is a primary goal of contrastive learning (CL) in cross-modal retrieval[1,2]. Our assumption follows this classical CL principle and extends it to a prototype-based formulation. Although CL encourages the model to pull semantically similar pairs together and push dissimilar pairs apart, prior studies[3,4], our Fig. 1, and the additional visualizations in Appendix Sec. C (Fig. 9 and Fig. 11) show that instance-level contrast alone often fails to produce sufficiently discriminative features.
>
> To address this limitation, we introduce PRL, a prototype-level contrastive framework that promotes a more structured embedding space by making representations within the same cluster more similar and compact, and representations across different clusters more separable.
>
> Following the core idea of CL, PRL offers two key advantages:
>
> 1. First, PRL forms video and text prototypes by clustering samples with similar semantics. We apply contrastive learning across prototypes to maintain semantic discrimination, and within prototypes to treat intra-cluster samples as hard negatives[5], which enhances fine-grained separability.
> 2. Second, it projects video and text prototypes into a shared semantic space and enforces alignment between each paired prototype. This leads to stronger cross-modal consistency than instance-level CL alone.
>
> Overall, our assumption and strategy acts as a regularization term to the CL, complementing rather than conflict it by addressing the IIC dilemma and producing more discriminative feature representations.
>
> **_Ref:_**
>
> [1] Deep Supervised Cross-modal Retrieval. CVPR 2019.
>
> [2] Rebalancing Contrastive Alignment with Learnable Semantic Gaps in Text-Video Retrieval. NeurIPS 2025.
>
> [3] On Learning Discriminative Features from Synthesized Data for Self-Supervised Fine-Grained Visual Recognition. ECCV 2024.
>
> [4] Which Features are Learnt by Contrastive Learning? On the Role of Simplicity Bias in Class Collapse and Feature Suppression. ICML 2023.
>
> [5] Hard Negative Sampling. ICLR 2021.
>
> ***
>
> > **Q-1**: How the category in the proposed motivation is defined? To what extent some concepts can be regarded as the same category, for example, whether two dogs with different breeds belong to the same category? Or whether the two verbs that describe the same action belong the same category? Since this lay the foundation for the motivation and the proposed method, a more rigorous (or quantitative) definition of it is expected beyond the conceptual illustration.
>
> **Response:** To illustrate motivation, we directly use predefined labels (20 categories for MSR-VTT, 200 for ActivityNet Captions, and 80 for MSCOCO) as the “categories.” For datasets without categorical labels (MSVD, DiDeMo, LSMDC, and Flickr30K), we randomly sample 1,000 videos per dataset and ask ten annotators to assign ten categories. The resulting category definitions are provided in **Appendix Sec. A, Fig. 7**.
>
> We agree that some concepts might be considered as the same category. It is hard even impossible to define a clear category boundary since visual scenes often contains multiple concepts. While such semantic overlap may cause IIC dilemma, *i.e.*, samples from genuinely different categories may be incorrectly retrieved due to strong superficial concepts (similar visual backgrounds or generic textual phrases), while samples within the same category are often dominated by these superficial concepts as well (*e.g.*, learning only a coarse “dog” representation while neglecting differences between breeds as exampled by reviewer). This hinders the model from learning the truly discriminative concepts.
>
> Therefore, our method avoids using explicit supervision and instead adopts unsupervised way, which allows prototypes to emerge directly based on the feature similarity. For a given sample *i* within a cluster, PRL regularizes confounded relations across clusters (*e.g.*, *drinking water* vs. *pouring water*, similar background for different sports). Once cross-cluster discriminability established, we regularize the representations within each cluster. The high similarity among intra-cluster samples makes them natural hard negatives. PRL preserves cluster compactness while driving the model to capture fine-grained distinctions (e.g., *two dogs with different breeds*).

---

> ### Author Response · Authors · 2025-11-27
> **Response to Reviewer CTri (3/3)**
>
> > **Q-2**: On some dataset, such as ActivityNet, the proposed method does not benefit but fails with CLIP3Clip, a failing case analyzation would be helpful.
>
> **Response:**  We have provided a case analyzation in **Appendix C with Fig.16** in the revised version. As observed, PRL yields successful retrieval reuslts given the text query. However, the benefits of PRL regularization become less pronounced for some complex and diverse scenes. Possible reasons are: (1) The optimal PRL hyperparameters vary across datasets and methods, while we use a fixed setting for all. (2) ActivityNet contains more complex scenarios than MSR-VTT (200 vs. 20 categories) and longer duration, making accurate matching more challenging.
>
> However, the failing cases with PRL still exhibit stronger semantic correspondence to the ground-truth videos than those produced by CLIP4Clip, showing its robustness.
>
> Moreover, suggested by your Q-3, we find that increase prototype number bringing better performance on ActivityNet dataset, *i.e.*, CLIP4Clip w/ PRL achieves 41.1 in R@1, which is superior than CLIP4Clip w/o PRL (40.5).
>
> ***
>
>
> > **Q-3**: How to determine the number of the prototypes across different methods and datasets? Why more count does not lead to more fine-grained representation learning and consistent improvement?
>
> **Response:** As discussed above, we just fix the number of prototypes to 16 across all methods and datasets for simplicity. Considering that optimal prototype counts may vary with dataset scale and distribution, we conducted experiments adjusting the number of prototypes across different methods and datasets.
>
> We have added this experiment in **Appendix Sec.B with Fig. 12**. As shown, larger datasets benefit from more prototypes due to richer semantic diversity. However, increasing the number of prototypes beyond a reasonable range can hurt performance, possible reason is that excessive prototypes split semantically similar instances across clusters thereby weaken the regularization effect.
>
> ***
>
> Finally, we would like to sincerely thank you for your careful review and valuable comments. We have revised our manuscript with discussions and experimental results according to your review. It is welcome if there are any further questions or suggestions.

---

### Meta-Review · Area_Chair_Q3kV · 2026-01-06

**Summary:**

The paper proposes Prototype-based Regularization Learning (PRL) to address the "Intra-Inter Conflict (IIC) dilemma" in text-video retrieval, where instances within the same category show high variance while instances from different categories are entangled. PRL is a plug-and-play, model-agnostic framework that employs clustering-based prototype mining and two regularization losses: Prototype Discriminating Loss (PDL) for semantic separation and Prototype Projection Loss (PPL) for cross-modal alignment. Reviewers acknowledged the strong motivation, the efficiency of the method (minimal overhead with zero learnable parameters), and its consistent performance gains across multiple baseline models and datasets.

**Reviewer Concerns:**

Despite the practical appeal of the plug-and-play module, several significant concerns were raised regarding benchmarking and technical depth. First, reviewers (F1QS, XRSn) pointed out the omission of several closely related prototype-based retrieval works (e.g., PAN, Adaptive Cross-modal Prototypes), which weakens the claim of novelty and superiority. Second, the technical implementation of mini-batch K-means clustering was questioned (CTri, QX4L, F1QS); specifically, the stability of clustering under biased mini-batch distributions, the handling of non-uniform batches, and the lack of a rigorous definition for "category" in an unsupervised setting. Finally, some reviewers found the performance gains on certain datasets (e.g., ActivityNet) to be marginal and questioned whether the IIC dilemma is a universal phenomenon or specific to certain architectures like CLIP.

**Reviewer Scores:**

The paper received scores of 6, 4, 4, and 4, indicating a borderline consensus that leans slightly toward rejection. While Reviewer XRSn (6) found the work effective, their confidence was low (2). Conversely, the three reviewers who gave a 4 (CTri, F1QS, QX4L) maintained high confidence (4 and 5) and identified specific gaps in the literature review and empirical validation of the clustering mechanism. If a full discussion period had occurred, the scores might have improved only if the authors provided a robust comparison with the missing prototype-based baselines and statistical evidence of clustering stability across different batch sizes. Without these critical additions, the majority of the reviewers remained hesitant to cross the acceptance threshold, leading to a recommendation of rejection.

---

### Decision · Program_Chairs · 2026-01-26

Reject